# RIM-BP2 primes synaptic vesicles *via* recruitment of Munc13-1 at hippocampal mossy fiber synapses

Marisa M Brockmann[1†], Marta Maglione[2,3,4†], Claudia G Willmes[5†], Alexander Stumpf[6], Boris A Bouazza[1], Laura M Velasquez[6], M Katharina Grauel[1], Prateep Beed[6], Martin Lehmann[3], Niclas Gimber[6], Jan Schmoranzer[4], Stephan J Sigrist[2,4,5*], Christian Rosenmund[1,4*], Dietmar Schmitz[4,5,6*]

[1]Institut für Neurophysiologie, Charité – Universitätsmedizin Berlin, corporate member of Freie Universität Berlin, Humboldt-Universität zu Berlin, and Berlin Institute of Health, Berlin, Germany; [2]Freie Universität Berlin, Institut für Biologie, Berlin, Germany; [3]Leibniz-Forschungsinstitut für Molekulare Pharmakologie (FMP), Berlin, Germany; [4]NeuroCure Cluster of Excellence, Berlin, Germany; [5]DZNE, German Center for Neurodegenerative Diseases, Berlin, Germany; [6]Neuroscience Research Center, Charité – Universitätsmedizin Berlin, corporate member of Freie Universität Berlin, Humboldt-Universität zu Berlin, and Berlin Institute of Health, Berlin, Germany

**\*For correspondence:**
stephan.sigrist@fu-berlin.de (SJS);
christian.rosenmund@charite.de (CR);
Dietmar.Schmitz@charite.de (DS)

[†]These authors contributed equally to this work

**Competing interests:** The authors declare that no competing interests exist.

**Abstract** All synapses require fusion-competent vesicles and coordinated $Ca^{2+}$-secretion coupling for neurotransmission, yet functional and anatomical properties are diverse across different synapse types. We show that the presynaptic protein RIM-BP2 has diversified functions in neurotransmitter release at different central murine synapses and thus contributes to synaptic diversity. At hippocampal pyramidal CA3-CA1 synapses, RIM-BP2 loss has a mild effect on neurotransmitter release, by only regulating $Ca^{2+}$-secretion coupling. However, at hippocampal mossy fiber synapses, RIM-BP2 has a substantial impact on neurotransmitter release by promoting vesicle docking/priming and vesicular release probability *via* stabilization of Munc13-1 at the active zone. We suggest that differences in the active zone organization may dictate the role a protein plays in synaptic transmission and that differences in active zone architecture is a major determinant factor in the functional diversity of synapses.
DOI: https://doi.org/10.7554/eLife.43243.001

## Introduction

Across all types of synapses, vesicle fusion is coordinated by an evolutionarily conserved set of vesicular and active zone proteins (*Südhof, 2012*). One hallmark of synapses is their functional heterogeneity: indeed, synapses can exhibit high or low transmission fidelity, and this diversity results in synapse-specific differences in response fluctuation and short-term plasticity (*Atwood and Karunanithi, 2002*; *O'Rourke et al., 2012*). In recent years, functional synaptic diversity has been found to be critical for routing and encoding sensory information within networks of neurons in the brain (*Chabrol et al., 2015*). Functional synaptic diversity has been observed both within and across brain regions, and it has been shown to play a significant role in temporal coding of multisensory integration and extraction of specific sensory features (*Atwood and Karunanithi, 2002*; *O'Rourke et al., 2012*; *Chabrol et al., 2015*). Still, the molecular origin of this heterogeneity is largely unknown, and analyses of genotype-phenotype differences across species and brain tissue have just started to uncover key molecular principles responsible for synaptic diversity, emphasizing the importance of

abundance and isoforms differences across species for synaptic diversity (*Rosenmund et al., 2002*; *Weston et al., 2011*; *Hu et al., 2013*; *Böhme et al., 2016*).

It is possible, but largely untested whether the active zone architecture, which is specialized throughout synapse types, is associated with distinct protein functions and thus contributes to synaptic diversity. Here, RIM-binding proteins (RIM-BPs) are particularly interesting, as their loss manifests in severe phenotypes in the *Drosophila* neuromuscular junction (NMJ) (*Liu et al., 2011*), but rather subtle phenotypes in small central murine synapses, the Calyx of Held or the ribbon synapse with only mild impairments in $Ca^{2+}$-channel-release site coupling (*Acuna et al., 2015*; *Grauel et al., 2016*; *Luo et al., 2017*; *Davydova et al., 2014*). *Drosophila* NMJ and small central synapses are considerable distinct in their anatomical, ultrastructural and, physiological properties (*Ackermann et al., 2015*).

To understand whether the RIM-BP2 phenotypes described so far are species or synapse type dependent, we chose to examine RIM-BP2 function at mouse hippocampal mossy fiber (MF) synapses, a mammalian synapse with distinct physiologically and anatomically properties. Notably, MF synapses strongly facilitate and possess multiple release sites (*Nicoll and Schmitz, 2005*).

Together, our laboratories previously published a detailed analysis on RIM-BP2 function at Schaffer collateral (SC; CA3-CA1) synapses using murine hippocampal autaptic neurons and acute brain slices (*Grauel et al., 2016*). Combining electrophysiological recordings and gSTED analysis, we concluded that RIM-BP2 mildly affects neurotransmissions by altering $Ca^{2+}$-channel-release site coupling and affecting synaptic release probability at these synapses.

Now, extended analysis of how RIM-BP2 impacts on synaptic integrity revealed that neurotransmission at MF synapses is severely impaired upon the loss of RIM-BP2, compared to that at SC synapses. Furthermore, we also show that RIM-BP2 loss leads to a defective stabilization of Munc13-1 clusters at the active zone specifically at MF synapses, but not at SC synapses, indicative of diversified functions of RIM-BP2 at these two synapse types. While at SC synapses RIM-BP2 maintains high fidelity coupling of $Ca^{2+}$-channels to release sites, at MF synapses RIM-BP2 is required to stabilize Munc13-1 clusters to ensure vesicle docking/priming. In addition, RIM-BP2 deletion alters vesicular release probability at MF terminals most probably *via* increased distances between $Ca^{2+}$-channels and release sites mapped by Munc13-1. Finally, our analysis of the active zone architecture revealed that RIM-BP2 and Munc13-1 clusters as well as RIM1 and Cav2.1 clusters are positioned at increased distances in MF synapses compared to that in CA3-CA1 synapses, demonstrating that these synapses utilize different architectural organizational principles.

## Results

### Distinct role of RIM-BP2 at hippocampal synapses

To probe the nature of diversity between central mammalian synapses, we examined the role of RIM-BP2 throughout the hippocampus. Immunostainings for RIM-BP2 in mouse hippocampal slices revealed RIM-BP2 expression in the whole hippocampal neuropil, with a strong labeling of the mossy-fiber layer band in the CA3 stratum lucidum (*Figure 1a*).

To analyze the functional impact of RIM-BP2 loss at different hippocampal synapses, we recorded field excitatory postsynaptic potentials (fEPSPs) in acute brain slices obtained from RIM-BP2 KO mice and wildtype (WT) littermates. Synaptic transmission was assessed at three different hippocampal synapses: the Mossy fiber – pyramidal cell synapse (MF-CA3), the Associational – commissural synapses (AC-CA3), and the SC synapse (CA3-CA1) (illustrated in *Figure 1a*). To ensure MF origin, we verified input sensitivity to group II metabotropic glutamate receptor (mGluR) agonist DCG IV (*Yoshino et al., 1996*). The ratio of fEPSP to presynaptic fiber volley (PFV) was drastically reduced when stimulating the MF pathway in RIM-BP2 deficient (KO) slices compared to that in WT slices (*Figure 1c*). This shows that neurotransmission is severely impaired upon loss of RIM-BP2 at MF synapses. In contrast, the ratio of fEPSP to presynaptic fiber volley (PFV) for associative commissural (AC)-fibers and Schaffer collaterals (SC), both representing small central synapses, were not affected by the loss of RIM-BP2 (*Figure 1b*). Thus, RIM-BP2 deletion specifically impairs neurotransmitter release at hippocampal MF synapses, compared to AC and SC synapses.

To further characterize the defect in neurotransmission at MF synapses upon loss of RIM-BP2, we analyzed frequency facilitation at MF synapses by applying a stimulus train of 1 Hz. Normalized

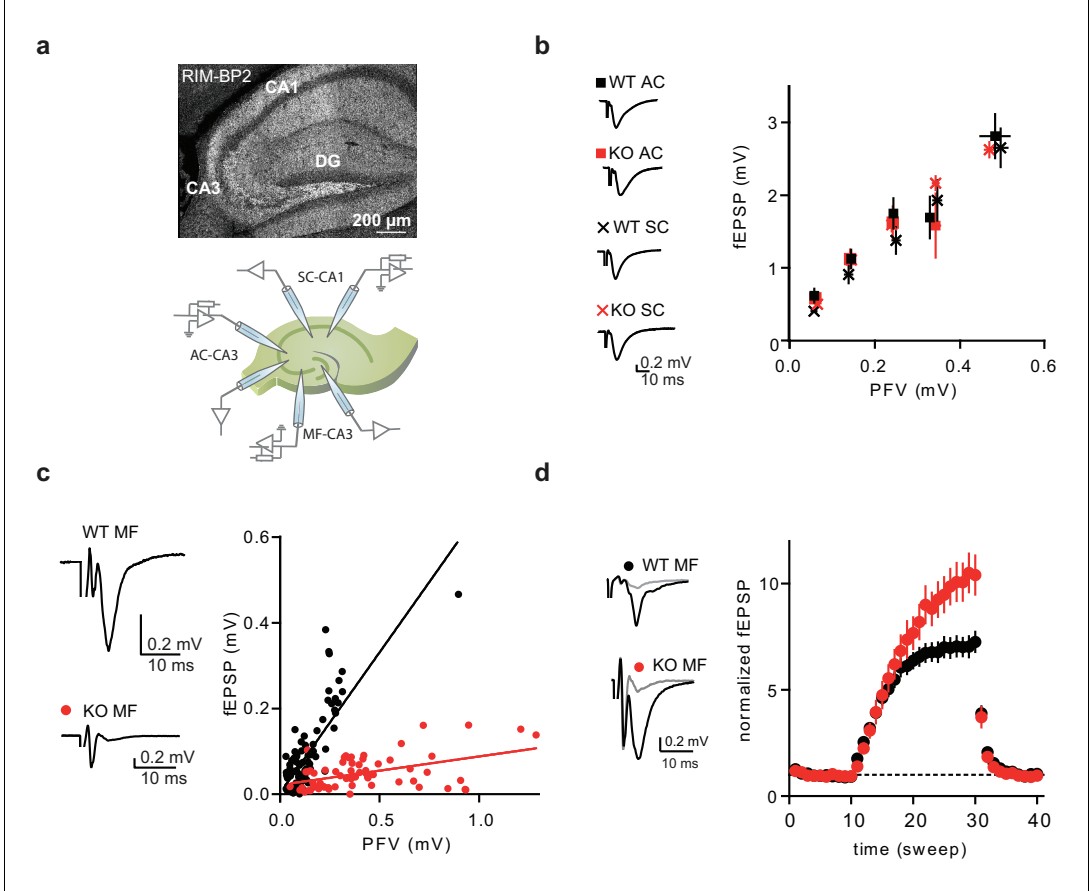

**Figure 1.** RIM-BP2 KO affects synaptic transmission specifically at MF synapses. (**a**) Immunostaining of RIM-PB2 in hippocampal brain slices (DG = dentate gyrus) and schematic illustration of recording configurations. (**b**) Input-output of synaptic transmission, plotted as PFV against fEPSP amplitude, of associative commissural (AC) and Schaffer collateral (SC) synapses showed no difference between RIM-BP2 WT and KO slices (AC: n(WT) =17 slices/6 animals; n(KO)=21 slices/6 animals) (SC: n(WT)=9 slices/3 animals; n(KO)=12 slices/3 animals). Sample traces show averages of 10 sweeps. Values represent mean ± SEM. (**c**) Input-output of synaptic transmission of MF synapses, plotted as PFV against fEPSP amplitude (MF: n(WT)=22 slices/7 animals; n(KO)=18 slices/7 animals). Sample traces show averages of 10 sweeps. (**d**) Frequency facilitation with 1 Hz stimulation of MF synapses (sweep 10–30). Sample traces show averages of five sweeps before (gray) and at the end of 1 Hz stimulation (black). For statistics please see *Figure 1—source data 1*.

DOI: https://doi.org/10.7554/eLife.43243.002

The following source data is available for figure 1:

**Source data 1.** Statistics of data presented in *Figure 1*.

DOI: https://doi.org/10.7554/eLife.43243.003

fEPSPs amplitudes were significantly increased in RIM-BP2 deficient synapses at the end of 1 Hz train stimulation, suggesting a role of RIM-BP2 in short-term plasticity at the MF synapse (*Figure 1d*).

In addition, we recorded in whole cell patch-clamp mode from CA3-pyramidal neurons in acute hippocampal slices and stimulated MF EPSCs. However, it was extremely difficult to find a quantifiable input of mossy fibers onto CA3 pyramids in the RIM-BP2 KOs, which was in sharp contrast to responses from WT animals. This strong phenotype made a comparative analysis of synaptic properties using whole cell recordings unfeasible.

## RIM-BP2 deletion does not alter Ca$^{2+}$-channel localization at the MF synapse

Given that RIM-BP2 contributes to high fidelity coupling of Ca$^{2+}$-channels and release apparatus in CA3-CA1 synapses (*Acuna et al., 2015*; *Grauel et al., 2016*), the disruption in synaptic transmission in the MF synapse may also arise from alterations in active zone organization.

Since MF terminals are unique in their morphology, we first assessed the active zone molecular architecture from WT mice. Therefore, we used triple-channel gSTED, with a lateral resolution of approximately 50 nm in all channels (*Grauel et al., 2016*). Due to limited capacity of antibodies combination, we could not combine specific markers to differentiate between MF postsynaptic partners in our immunostainings. We thereby refer to MF terminals or synapses in our analysis. In order to define putative differences in the active zone architecture at MF and CA3-CA1 synapses, we utilized super-resolution STED-microscopy based on detection of major active zone proteins intensities, here referred to as protein clusters. Quantification of the localization of these protein clusters was mainly performed by counting and measuring distances between intensities of these marker proteins. We compared the distribution of RIM-BP2 to the active zone markers Munc13-1 and Bassoon as a proxy for active zone organization. Protein clusters were determined by peak intensities, after image thresholding and watershed segmentation. We analyzed the $k$ nearest neighbor distances ($d_k$) between clusters formed by these proteins by measuring Euclidean distances between the centers of each cluster, using a semi-automated analysis described previously (*Grauel et al., 2016*) (*Figure 2a–c*). Since the $k$ nearest neighbor distance analysis was performed in parallel on the same processed brain slice preparations as our previously published gSTED experiments at CA3-CA1 synapses, we were able to compare protein cluster distances between both synapse types. The average distance of the closest RIM-BP2 cluster relative to a given Bassoon cluster at MF synapses was comparable to CA3-CA1 synapses. However, the closest Munc13-1 cluster in the MF synapse was 51% further away from a RIM-BP2 cluster ($174 \pm 20$ nm, *Figure 2c*) than what we previously observed at CA3-CA1 synapses ($115 \pm 5$ nm) (*Grauel et al., 2016*). Our gSTED analysis does not allow the differentiation between intra- and inter- active zone protein clusters. However, ultrastructural quantifications of the MF active zone size (0.12 μm2) (*Rollenhagen et al., 2007*), are consistent with four RIM-BP2, two Bassoon, and three Munc13-1 clusters per active zone (*Figure 2c*). Regardless of this semiquantitative analysis, the difference in Munc13-1/RBP cluster distances is indicative for distinct active zone organization between MF and CA3-CA1 synapses.

We next asked whether changes in protein cluster distribution might account for the impaired release in RIM-BP2 deficient MF synapses. First, in accordance to the phenotype observed at the CA3-CA1 synapse (11,12d), we determined the position of clusters formed by the P/Q type $Ca^{2+}$-channel subunit Cav2.1 in relation to clusters formed by the active zone protein RIM1, and the postsynaptic scaffold Homer1 in RIM-BP2 WT and KO MF synapses. Surprisingly, neither the number, the ratio of protein clusters nor the distance of RIM1 or Homer1 clusters to a given Cav2.1 cluster were altered upon the loss of RIM-BP2 (*Figure 2d–f*). As for RIM-BP2, Munc13-1, and Bassoon we computed the putative number of clusters formed by Cav2.1 and RIM1 that would fit in an average MF active zone (*Figure 2g*). According to the threshold applied, we detected in average three Cav2.1 clusters and one RIM1 clusters per single active zone in WT mice (*Figure 2h*). We further analyzed the cluster number and $k$ nearest neighbor distances of RIM1, as main interacting scaffold protein of RIM-BP2. The cluster number of RIM1 as well as the nearest neighbor distance between Homer1 and RIM1 clusters was unaltered in RIM-BP2 deficient MF synapses (*Figure 2g,h*). Although we observed a higher variability in the RIM1/Cav2.1 cluster ratio in RIM-BP2 KO mice (*Figure 2e,f*), our data suggest that the loss of RIM-BP2 does not grossly alter RIM1 cluster number or localization at MF terminals.

To probe again for differential organizations of MF and CA3-CA1 synapses, we compared in simultaneously acquired images from the same brain slices (*Grauel et al., 2016*), the distance between RIM1 and Cav2.1 clusters in both synapses. Interestingly, we found a 35% larger distance between RIM1 and Cav2.1 in MF synapses, again suggesting a differential nanoarchitecture of MF active zones.

Our unbiased $k$ nearest neighbor analysis measures the distance of thousands of neighboring protein clusters found in one image. However, it does not allow us to discriminate protein clusters within or between nearby active zones unequivocally. To retrieve a more direct determination of distances within the active zone, we performed line profile measurements of peak-to-peak distances between protein clusters in selected active zones that are defined to be opposite to the postsynaptic marker Homer1. We analyzed only well-defined synapses in side or planar view. As shown with our $k$ nearest neighbor analysis, line profile measurements did not reveal a significant change in the distance between two adjacent Cav2.1 clusters comparing RIM-BP2 WT and KO MF synapses (*Figure 2—figure supplement 1*), indicative of unaltered P/Q type $Ca^{2+}$-channel localization within a given MF

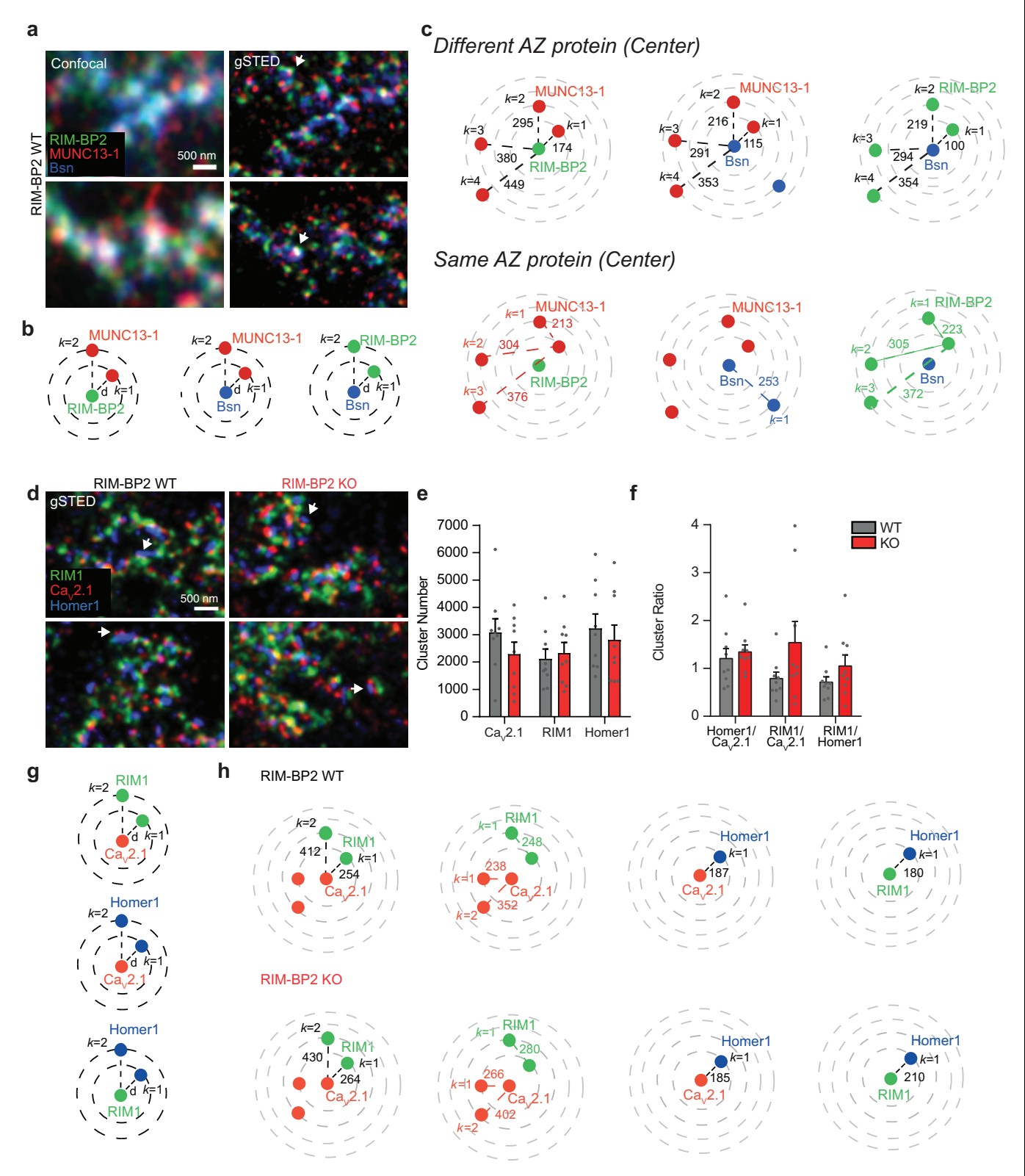

**Figure 2.** RIM-BP2 deletion does not alter the localization of Ca$_V$2.1 clusters relative to the active zone protein RIM1 and the postsynaptic marker Homer1 at MF synapses. (a) Confocal (left) and gSTED (right) images of RIM-BP2, Munc13-1 and Bassoon (Bsn) at the active zone (AZ) of WT MF boutons (MFBs) in situ. Arrows indicate synapses in side view. (b) Example of *k* nearest neighbor distance analysis of protein clusters at MF synapses. Following image thresholding and Watershed segmentation, X and Y coordinates of each segmented cluster identified were retrieved and Euclidean

*Figure 2 continued on next page*

*Figure 2 continued*

distances of for example Munc13-1 clusters relative to a given RIM-BP2 cluster calculated with a custom-written MATLAB script. Several hundreds to thousands of clusters per image were analyzed and values averaged per animal (n = 6). (c) Upper, mean *k* nearest neighbor distances for Munc13-1 clusters relative to a given RIM-BP2 (left) or Bassoon (middle) cluster and for RIM-BP2 clusters relative to a given Bassoon cluster (right). Lower, mean *k* nearest neighbor distances for Munc13-1 clusters relative to Munc13-1 itself as center (left), for Bassoon clusters relative to itself (middle) and RIM-BP2 clusters relative to itself (right). Based on ultrastructural studies of MF AZ size, estimating an AZ diameter of 391 nm, at WT MFBs we detected at least three Munc13-1 clusters, two Bassoon clusters and four RIM-BP2 clusters within a single AZ, having a $d_k$ <391 nm.(d) gSTED images of Ca$_V$2.1, RIM1 and Homer1 clusters at MFBs of RIM-BP2 WT and KO brain slices. Arrows indicate synapses with two Cav2.1 clusters apposed to a single Homer1 cluster. (e) Average number of Cav2.1, RIM1 and Homer1 clusters found at MFBs and cluster ratio per each RIM-BP2 WT (n = 9) and KO (n = 9) mouse analyzed (f). No significant differences were observed between the two groups. (g) Example of *k* nearest neighbor distance analysis of protein clusters at MF synapses. Several hundreds to thousands of clusters per image were analyzed and values averaged per animal. (h) *k* nearest neighbor distances of the first and second closest RIM1 *k* neighbor (*k* = 1, *k* = 2) relative to a given Cav2.1 (first left), no significant differences were observed between RIM-BP2 WT and KO mice. No significant differences were observed also for the mean *k* nearest neighbor distance at which Cav2.1 clusters are located relative to a given Cav2.1 (second left). Based on ultrastructural studies of MF AZ size, estimating an AZ diameter of 391 nm, at WT MFBs we detected one RIM1 cluster and three Cav2.1 clusters per single active zone, having a $d_k$ <391 nm. No significant difference was observed for Cav2.1 and RIM1 localization in relation to Homer1 (third left and first right, respectively). Values represent mean ± SEM. For statistics please see *Figure 2—source data 1*.

DOI: https://doi.org/10.7554/eLife.43243.004

The following source data and figure supplements are available for figure 2:

**Source data 1.** Statistics of data presented in *Figure 2*.

DOI: https://doi.org/10.7554/eLife.43243.007

**Figure supplement 1.** RIM-BP2 deletion does not alter P/Q type Ca$^{2+}$-channels localization at MF synapses.

DOI: https://doi.org/10.7554/eLife.43243.005

**Figure supplement 1—source data 1.** Statistics of data presented in *Figure 2—figure supplement 1*.

DOI: https://doi.org/10.7554/eLife.43243.006

active zone upon loss of RIM-BP2. Our line profile measurements retrieved a distance between two Cav2.1 clusters of 184 ± 6 nm, a distance shorter than the one obtained with our semi-automated analysis (238 ± 23 nm), possibly indicating a subpopulation of synapses selected or mapping of clusters belonging to neighboring active zones, respectively.

## RIM-BP2 stabilizes Munc13-1 protein clusters at MF synapses

We next compared the relative localization and abundance of Munc13-1 clusters and their distance to Ca$^{2+}$-channels in RIM-BP2 WT and KO brain slices. In these experiments we identified MF synapses by their expression of the vesicular ZnT3, enriched at these synapses (*Wenzel et al., 1997*). Interestingly, we found a drastic reduction in the number of Munc13-1 clusters in RIM-BP2 deficient synapses, accompanied by a decrease in Munc13-1 clusters at specific distances from a given Cav2.1 clusters (*Figure 3a–e*). Therefore, in RIM-BP2 deficient MF active zones we detect less Munc13-1 clusters, which also were located at an increased distance to Cav2.1 clusters compared to WT MF synapses (*Figure 3f,g*). In contrast, the Munc13-1 cluster number and distribution at CA3-CA1 synapses were unaltered upon RIM-BP2 loss (*Figure 3h–m*), indicating that the observed reduction in the Munc13-1 cluster number is specific for MF synapses. To ensure the analysis of intra-active zone clusters, we performed line profile measurements assessing the peak intensity distance between Munc13-1 and Cav2.1 clusters in RIM-BP2 WT and KO MF synapses. Peak-to-peak distance measurements revealed a significantly increased distance of Munc13-1 clusters relative to Cav2.1 clusters in RIM-BP2 deficient MF synapses compared to WT (*Figure 3—figure supplement 1*), supporting our results from the semi-automated *k* nearest neighbor analysis.

To further validate our results on Munc13-1 localization within a single active zone, we analyzed by line profile measurements inter-cluster distances of two Munc13-1 clusters opposed to Homer1 at MF and CA3-CA1 terminals of RIM-BP2 WT and KO mice. Confirming our nearest neighbor analysis, the peak-to-peak distance between two Munc13-1 clusters increased at RIM-BP2 KO MF active zones, in contrast to CA3-CA1 synapses (*Figure 3—figure supplement 2*). Notably, as shown for Cav2.1 inter-cluster measurements, the distances between Munc13-1 and Cav2.1 clusters or inter-Munc13-1 clusters retrieved by line profile measurements were shorter than the one retrieved with our semi-automated analysis, possibly indicating a subpopulation of synapses selected or mapping of clusters belonging to neighboring active zones, respectively.

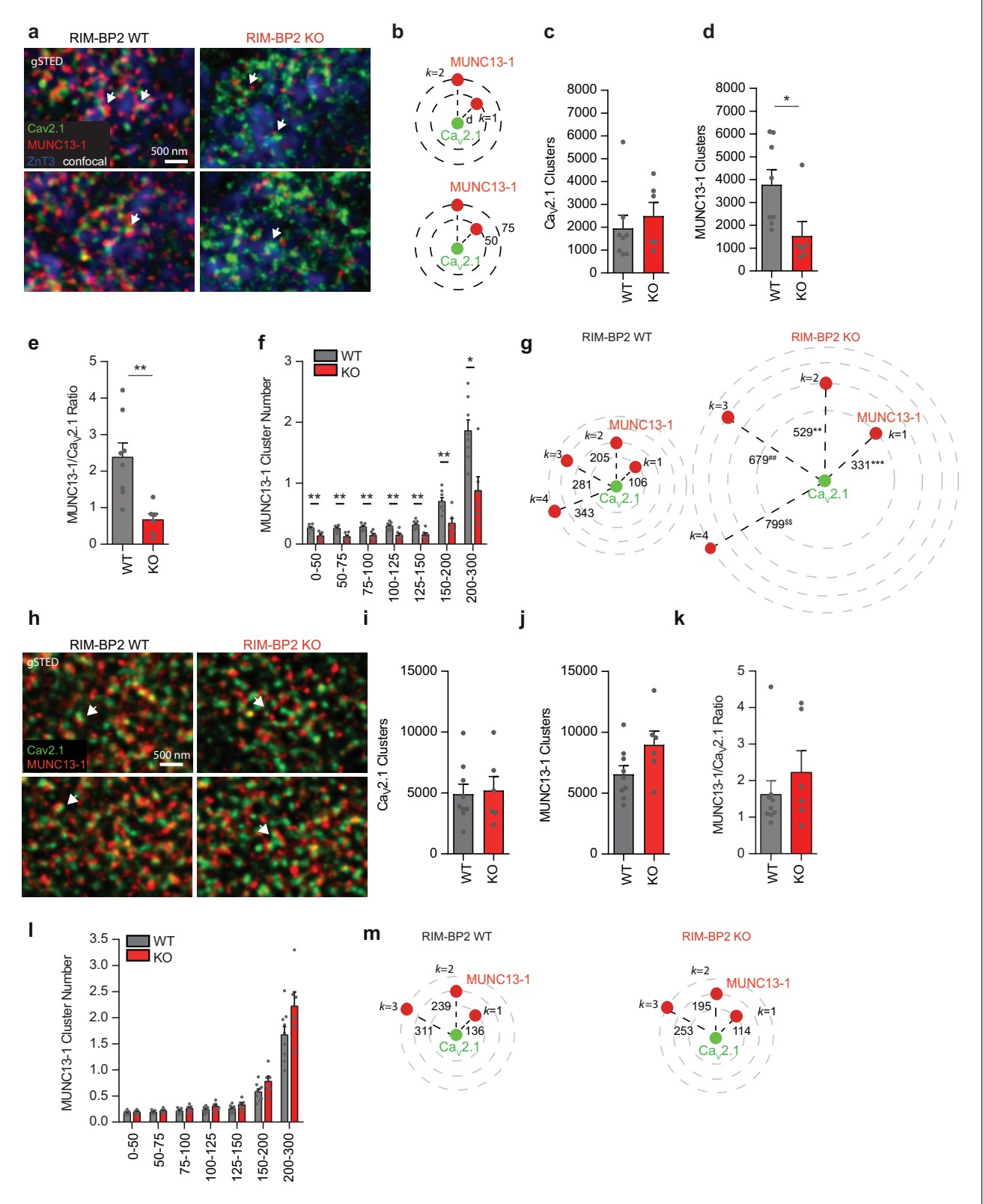

**Figure 3.** Loss of RIM-BP2 specifically reduces Munc13-1 levels at MF synapses but not at CA3-CA1 synapses. (a) Representative gSTED images of Ca$_V$2.1 and Munc13-1 clusters at MF boutons (MFB) identified by ZnT3 expression (confocal) in RIM-BP2 WT and KO brain sections. Arrows indicate Munc13-1 clusters nearby Ca$_V$2.1 clusters. (b) Example of $k$ nearest neighbor distance analysis of Munc13-1 clusters relative to a given Cav2.1 cluster at MF synapses. Following image thresholding and Watershed segmentation, X and Y coordinates of each segmented cluster identified were retrieved

*Figure 3 continued*

and Euclidean distances of for example Munc13-1 clusters relative to a given Cav2.1 cluster calculated with a custom-written MATLAB script (b, upper). Similarly, we retrieved the number of for example Munc13-1 clusters found at specific distance intervals (nm) from a given Cav2.1 cluster (b, lower). Several hundreds to thousands of clusters per image were analyzed and values averaged per animal. (c) Average number of Cav2.1 clusters within the ZnT3 +area found per each RIM-BP2 WT (n = 8) and KO (n = 6) animal analyzed. ZnT3 was used as marker to identify MF synapses. (d) Average number of Munc13-1 clusters within the ZnT3 +area found per each RIM-BP2 WT and KO animal analyzed (e) Ratio of Munc13-1 clusters/Cav2.1 clusters in RIM-BP2 KO and WT mice. (f) The number of Munc13-1 clusters at determined distance intervals (nm) from a given Cav2.1 cluster decreased significantly at all distances analyzed in RIM-BP2 KO, while the distance of the first closest *k* neighbor (*k* = 1; g) up to the fourth (*k* = 4) significantly increased. (h) Representative gSTED images of Cav2.1 and Munc13-1 clusters at CA3-CA1 synapses in RIM-BP2 WT and KO brain sections. Arrows indicate Munc13-1 clusters adjacent to Cav2.1 clusters. (i) Average number of Cav2.1 clusters and Munc13-1 clusters (j) found within the field of view at CA3-CA1 synapses in RIM-BP2 KO (n = 6) and WT (n = 9) mice (k) Ratio of Munc13-1 clusters/Cav2.1 clusters at CA3-CA1 synapses. (l) At CA3-CA1 synapses, loss of RIM-BP2 does not significantly alter either the number of Munc13-1 clusters at determined distance intervals (nm) from a given Cav2.1 cluster or the distance at which the first closest *k* neighbor (*k* = 1, (m) is found. Values represent mean ± SEM. *p<0.05, **p<0.01. For statistics please see *Figure 3—source data 1*.

DOI: https://doi.org/10.7554/eLife.43243.008

The following source data and figure supplements are available for figure 3:

**Source data 1.** Statistics of data presented in *Figure 3*.

DOI: https://doi.org/10.7554/eLife.43243.017

**Figure supplement 1.** Loss of RIM-BP2 leads to an increased distance between Munc13-1 and Cav2.1 clusters at MF synapses.

DOI: https://doi.org/10.7554/eLife.43243.009

**Figure supplement 1—source data 1.** Statistics of data presented in *Figure 3—figure supplement 1*.

DOI: https://doi.org/10.7554/eLife.43243.010

**Figure supplement 2.** Loss of RIM-BP2 leads to an increased inter-Munc13-1 clusters distance specifically at MF synapses.

DOI: https://doi.org/10.7554/eLife.43243.011

**Figure supplement 2—source data 1.** Statistics of data presented in *Figure 3—figure supplement 2*.

DOI: https://doi.org/10.7554/eLife.43243.012

**Figure supplement 3.** RIM-BP2 deletion results in significantly reduced Munc13-1 levels at MF synapses.

DOI: https://doi.org/10.7554/eLife.43243.013

**Figure supplement 3—source data 1.** Statistics of data presented in *Figure 3—figure supplement 3*.

DOI: https://doi.org/10.7554/eLife.43243.014

**Figure supplement 4.** Loss of RIM-BP2 does not alter Munc13-2 levels at both MF-CA3 and CA3-CA1 synapses.

DOI: https://doi.org/10.7554/eLife.43243.015

**Figure supplement 4—source data 1.** Statistics of data presented in *Figure 3—figure supplement 4*.

DOI: https://doi.org/10.7554/eLife.43243.016

Altogether, our gSTED analysis demonstrates that RIM-BP2 is essential in stabilizing Munc13-1 protein clusters specifically at MF synapses.

To analyze if the loss of RIM-BP2 synapse specifically affects protein clustering or total protein amount, we examined protein levels of Munc13-1 at SC and MF synapses. Munc13-1 intensity levels were quantified on confocal images acquired in parallel to corresponding gSTED images. Deletion of RIM-BP2 resulted in a significant reduction of Munc13-1 intensity levels at MF synapses but not at CA3-CA1 terminals (*Figure 3—figure supplement 3*).

We also analyzed the relative distribution and abundance of Munc13-2 clusters relative to a given Cav2.1 cluster at both MF and CA3-CA1 synapses. In both synapses, loss of RIM-BP2 neither altered the number of protein clusters nor the distribution of Munc13-2 clusters relative to Cav2.1 channels (*Figure 3—figure supplement 4*), showing that the observed electrophysiological phenotype is not due to Munc13-2 loss (*Breustedt et al., 2010*).

## RIM-BP2 docks synaptic vesicles *via* the specific recruitment of Munc13-1 at MF synapses

The primary role of Munc13-1 is to dock and prime synaptic vesicles (SV) at the active zone (*Imig et al., 2014*; *Augustin et al., 1999*). To explore whether the reduction in Munc13-1 clusters at RIM-BP2 deficient MF synapse might result in a decrease of vesicle docking, we analyzed their ultrastructural anatomy by electron microscopy (*Figure 4a*). We made use of high-pressure freezing fixation followed by freeze substitution, since conventional chemical fixation cannot reveal the Munc13-1 dependent loss in SV docking (*Varoqueaux et al., 2002*; *Camacho et al., 2017*). High-pressure

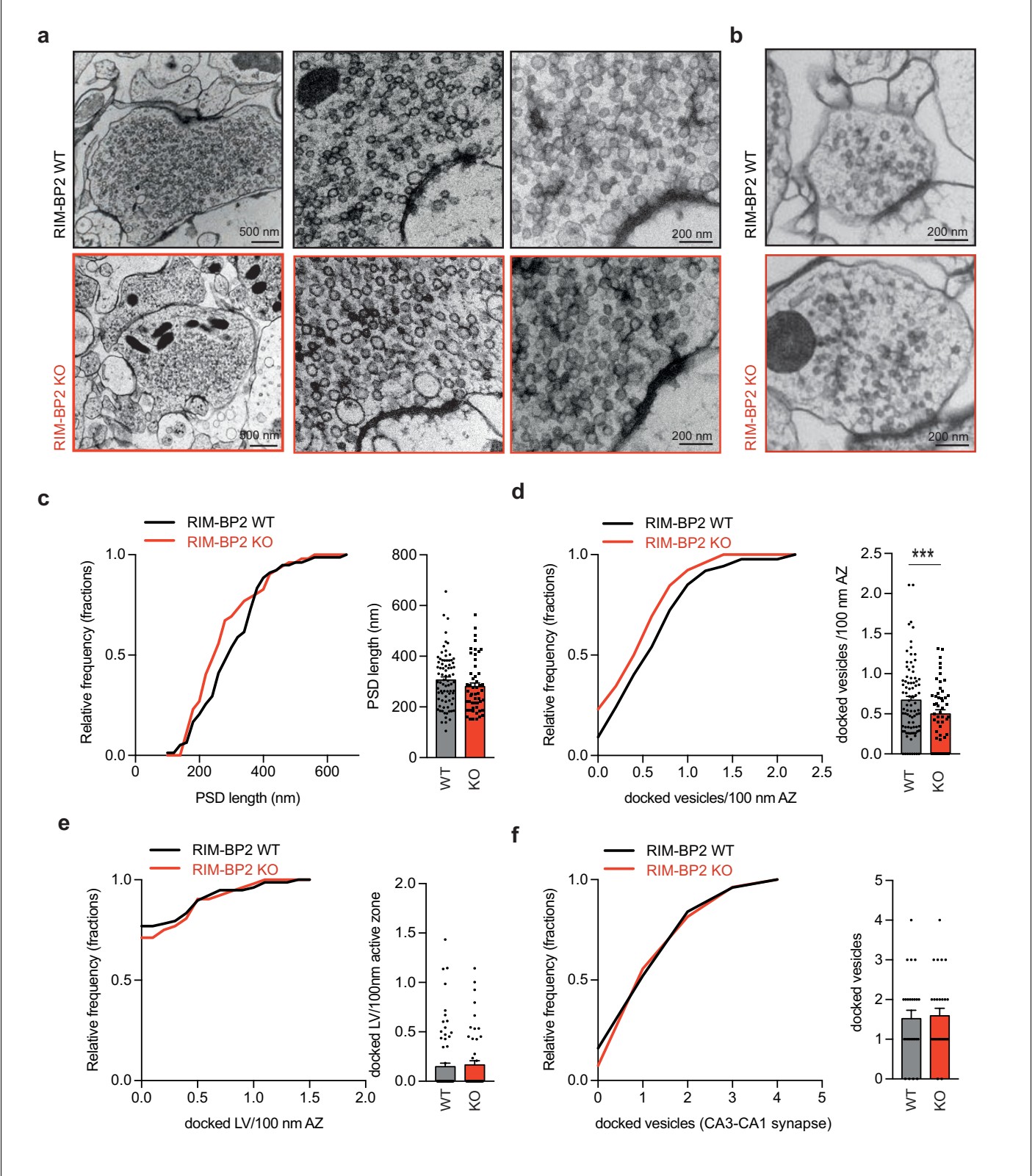

**Figure 4.** Loss of RIM-BP2 specifically affects vesicle docking at MF synapses. Representative EM images of MF synapses from acute hippocampal slices obtained from RIM-BP2 KO (black: three animals/each2-3 slices/78 active zones) and WT (red: three animals/2–3 slices each/52 active zones) mice. (b) Representative EM images of CA1 synapses from acute hippocampal slices obtained from RIM-PB2 KO (black: three animals/each2slices/27 active zones) and WT (red: three animals/2 slices each/25 active zones) mice. (c) Frequency distribution and bar graph show no difference in the size in the

*Figure 4 continued on next page*

*Figure 4 continued*

post-synaptic density (PSD) in MF active zones from WT or RIM-BP2 KO slices. (**d**) Frequency distribution and bar graph show a reduction of docked vesicles per 100 nm of the active zone at RIM-BP2 KO MF synapse compared to WT MF synapses. (**e**) Frequency distribution and bar graph depict no difference in the number of docked large vesicles (LV) (vesicle diameter >70 nm) (**f**) Frequency distribution and bar graph of docked vesicles at CA3-CA1 synapses show no difference between WT and RIM-BP2 KO. Values represent mean ± SEM. *p<0.05, **p<0.01, ***p<0.001. For statistics please see *Figure 4—source data 1*.

DOI: https://doi.org/10.7554/eLife.43243.018

The following source data is available for figure 4:

**Source data 1.** Statistics of data presented in *Figure 4*.

DOI: https://doi.org/10.7554/eLife.43243.019

freezing of acute hippocampal slices (4–6 weeks) show a ~ 25% reduction in docked SVs in RIM-BP2 deficient MF synapses compared to WT MF synapses (*Figure 4c*). The size of the post-synaptic density (PSD) was unaltered upon the loss of RIM-BP2 indicating that RIM-BP2 does not affect trans-synaptic interactions at MF synapses (*Figure 4b*). To evaluate whether the deletion of RIM-BP2 in acute hippocampal slices also affects SV docking at SC synapses we analyzed SV distribution at small central synapses in the CA1 area in parallel. However as expected from previous results in culture (*Grauel et al., 2016*), the number of docked SVs in CA1 synapses was unaltered in RIM-BP2 deficient synapses compared to synapses from WT slices (*Figure 4b,f*).

MF synapses also contain a small but distinct fraction of large clear vesicles (>70 nm diameter; *Henze et al., 2002*; *Rollenhagen and Lübke, 2010*), which generate mEPSC >100 pA upon fusion (*Henze et al., 2002*). The number of large docked vesicles per active zone was however not affected by the deletion of RIM-BP2 (*Figure 4e*).

Thus, in contrast to CA3-CA1 synapses, MF synapses require RIM-BP2 dependent stabilization of Munc13-1 at the active zone to dock synaptic vesicles.

## Loss of RIM-BP2 impairs vesicle priming and release efficiency in granule autaptic neurons

To examine the functional impact of RIM-BP2 loss in MF synapses, we prepared autaptic cultures of hippocampal granule cells that form synaptic contacts only with themselves and therefore allow the quantitative analysis of synaptic input-output properties. It is important to note that cultured granule cells form MF like boutons as assessed by EM analysis (*Figure 5—figure supplement 1*) and are sensitive to the application of DCG IV (*Figure 5a,b* and *Figure 5—figure supplement 1*). Therefore, granule autaptic neurons exhibit key aspects of hippocampal granule cell identity. Interestingly, since autapses just form synapses with themselves, their specialized protein expression and presynaptic ultrastructure are likely to be intrinsically encoded and mostly independent of the postsynaptic target.

We first tested whether autaptic granule neurons also show a reduction in neurotransmitter release upon RIM-BP2 loss and thus whether they can be used as a model system to study synapse diversity. Consistent with the findings from hippocampal field recordings (*Figure 1*), evoked excitatory postsynaptic currents (EPSCs) were severely impaired in RIM-BP2 KO granule cell autapses as compared to that in WT autapses (*Figure 5a,b*). Rescue of RIM-BP2 deficiency by lentiviral expression in RIM-BP2 KO neurons completely restored synaptic transmission, confirming the specificity of RIM-BP2 function at these synapses (*Figure 5a,b*). To examine the origin of the impaired evoked response, we first probed vesicle priming by measuring the readily releasable pool (RRP) *via* hypertonic sucrose solution application. In line with the finding that RIM-BP2 KO MF synapses had only half the number of docked vesicles compared to that in WT neurons (*Figure 4*), the size of the RRP was significantly reduced (*Figure 5c,d*). The frequency and amplitude of $Ca^{2+}$-independent release events as measured by recording spontaneous miniature EPSCs (mEPSCs) were not significantly altered upon RIM-BP2 loss (*Figure 5h* and *Figure 5—figure supplement 2a*). We next assessed vesicular release probability Pvr, as the ratio of EPSC and RRP charge, and RIM-BP2 KO and WT granule autapses were not significantly different (*Figure 5d*). However, utilizing paired-pulse ratio (PPR, 25 ms interstimulus interval), we found that RIM-BP2 KO neurons displayed enhanced facilitation compared to WT neurons, which could be rescued by re-expression of RIM-BP2 (*Figure 5e,f*).

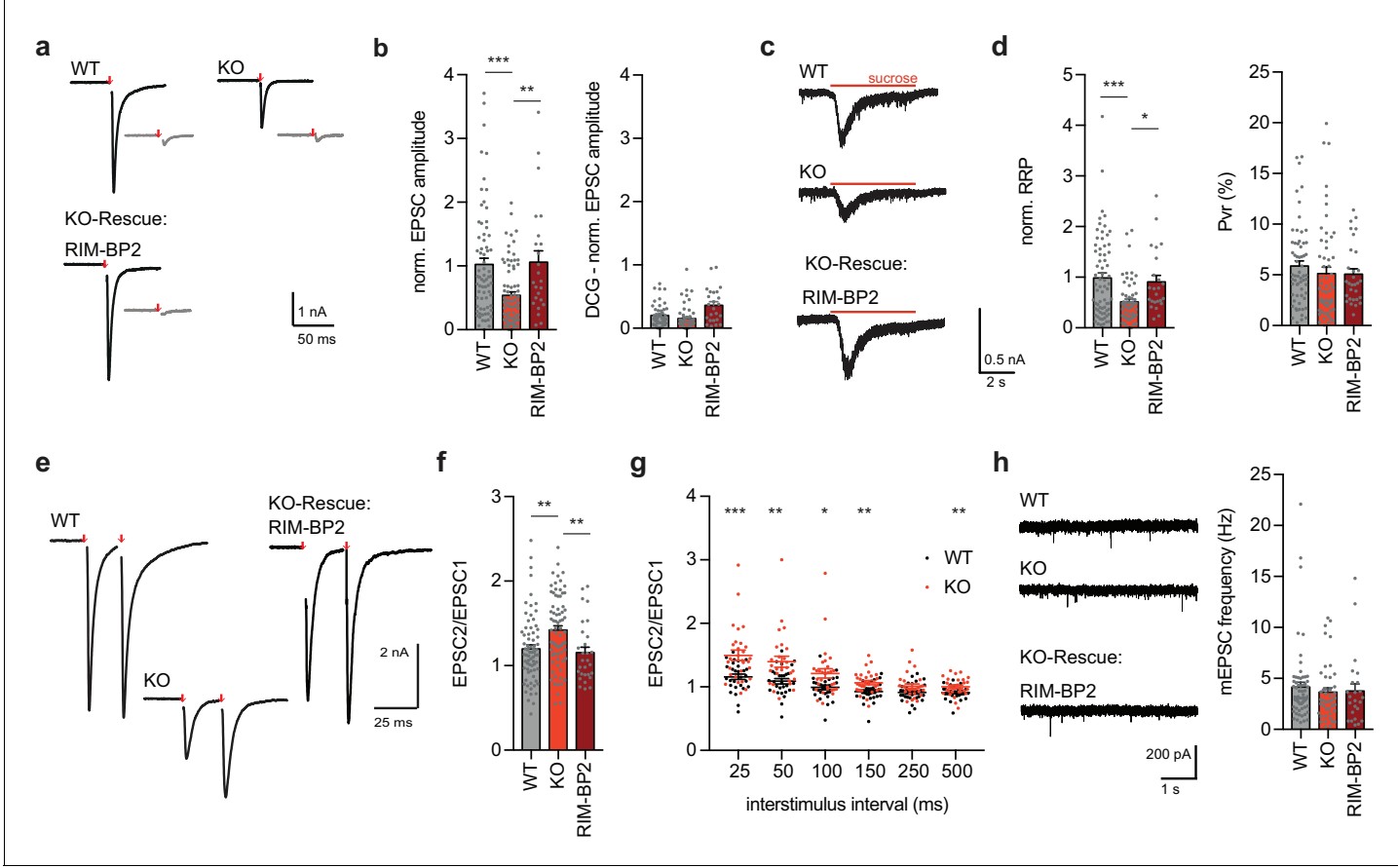

**Figure 5.** RIM-BP2 KO affects synaptic transmission at granule autaptic neurons. (**a**) Sample traces of evoked EPSC amplitudes before (black) and after DCG IV application (gray) for RIM-BP2 WT and KO neurons. RIM-BP2 KO neurons were rescued by lentiviral transduction of RIM-BP2. (Number of experiments (cells/cultures); EPSC WT (70/4); KO (78/4); RIM-BP2 (24/3)) (**b**) Summary graphs of normalized EPSC amplitudes evoked by 2 ms depolarization (red arrow). (**c**) Sample traces and (**d**) summary graphs of normalized RRP responses elicited by a 5 s application of 500 mM sucrose. Summary graph of the $P_{VR}$ calculated as the ratio of the EPSC charge and the RRP charge. (Sucrose WT (69/4); KO (59/4); RIM-BP2 (24/3)) .(**e**) Sample traces of evoked EPSC amplitudes with an interstimulus interval of 25 ms. (**f**) Summary graph of paired-pulse ratio (PPR) of RIM-BP2 WT, KO and RIM-BP2 rescued autaptic granule neurons. (PPR WT (70/4); KO (74/4); RIM-BP2 (24/3) (**g**) Summary graph of Paired-Pulse-Ratio (PPR) of granule cells with different inter-stimulus intervals of RIM-BP2 WT and KO granule autapses (PPR WT (28/2), PPR KO (29/2)). (**h**) Sample traces of miniature EPSCs (mEPSCs) and summary graph of mEPSC frequencies. (mEPSC WT (63/4); KO (43/4); RIM-BP2 (24/3)). Values represent mean ± SEM. *p<0.05, **p<0.01, ***p<0.001. For statistics please see *Figure 5—source data 1*.

DOI: https://doi.org/10.7554/eLife.43243.020

The following source data and figure supplements are available for figure 5:

**Source data 1.** Statistics of data presented in *Figure 5*.

DOI: https://doi.org/10.7554/eLife.43243.024

**Figure supplement 1.** Characterization of autaptic granule cells.

DOI: https://doi.org/10.7554/eLife.43243.021

**Figure supplement 2.** Loss of RIM-BP2 increases synaptic facilitation in autaptic granule neurons.

DOI: https://doi.org/10.7554/eLife.43243.022

**Figure supplement 2—source data 1.** Statistics of data presented in *Figure 5—figure supplement 2*.

DOI: https://doi.org/10.7554/eLife.43243.023

We assessed PPR in more detail by extending interstimulus intervals to up to 500 ms. At all but one interstimulus intervals tested (250 ms), PPF was increased (*Figure 5g*). Also, EPSCs evoked by 10 Hz action potential trains displayed less depression over the 5 s train duration in RIM-BP2 KO granule cell autapses compared to WT or rescue groups (*Figure 5—figure supplement 2b,c*). Thus, loss of RIM-BP2 leads to impaired hippocampal granule cell output, due to changes in both vesicle docking/priming and due to changes in release during repeated stimuli.

## RIM-BP2 primes synaptic vesicles *via* Munc13-1 in granule autaptic neurons

Our gSTED imaging experiments revealed a decrease in presynaptic Munc13-1 clusters upon the loss of RIM-BP2 (*Figure 3*). In small central synapses, priming is attained by an interaction of Munc13 and RIM *via* the Munc13 $C_2A$ domain, which can be mimicked by the constitutively monomeric mutant Munc13-1 K32E lacking Munc13-1 homodimerization (*Deng et al., 2011*). To test whether vesicle priming can be rescued by restoring Munc13-1 function in RIM-BP deficient MF synapses, we lentivirally transduced Munc13-1 WT (M13$^{WT}$) or the constitutively monomeric Munc13-1 mutant (M13$^{K32E}$) (*Camacho et al., 2017*) in granule autapses. Remarkably, Munc13-1 K32E expression in RIM-BP2 KO granule neurons sufficed to rescue the RRP, whereas Munc13-1 WT was not sufficient to rescue vesicle priming upon RIM-BP2 deletion (*Figure 6d*). In line with the RRP, evoked release could not be rescued by Munc13-1 WT, but required the constitutive active form of Munc13-1 K32E (*Figure 6a,b*). Again, the Pvr was neither altered upon RIM-BP2 deletion nor by the rescue of Munc13-1 WT or Munc13-1 K32E (*Figure 6d*).

These results suggest that in MF synapses, unlike hippocampal pyramidal neuron synapses, the recruitment or stabilization of active monomeric Munc13-1 is RIM-BP2 dependent.

## Discussion

Chemical synapses have been highly diversified by evolution in their molecular composition, ultra-structure, and consequently function. In the last decades, presynaptic diversity was studied in synapses that exhibit ultrastructural differences, such as the T-bar structure in the neuromuscular junction of *Drosophila melanogaster* or the ribbon synapse of vertebrate photoreceptor cells (*Ackermann et al., 2015*). Ultrastructural diversity is often associated with the expression of specialized synaptic organizers, like Bruchpilot or RIBEYE, which shape presynaptic structure and function (*Ackermann et al., 2015*; *Wagh et al., 2006*; *Schmitz et al., 2000*). More recently, the heterogeneity of distinct synapses came into focus, since many central synapses seemingly express similar pre-synaptic proteins but still show distinct release probabilities and $Ca^{2+}$-secretion coupling. Synaptic heterogeneity might be achieved by several mechanisms, including variation in the abundance of single proteins or expression of different protein isoforms. Notably, a contribution of the exact nano-scale arrangement of active zone proteins to synaptic diversity has also been discussed (*Atwood and Karunanithi, 2002*; *Nusser, 2018*).

To understand the function of a given protein in neurotransmitter release, one can compile the results from protein knockouts in diverse synapses and extract a universal function. Some highly conserved proteins, like Munc13-1, are essential for neurotransmitter release in a variety of synapses, since they exclusively conduct one specific function, in this case vesicle docking and priming (*Augustin et al., 1999*; *Varoqueaux et al., 2002*; *Weimer et al., 2006*; *Aravamudan et al., 1999*). However, some other presynaptic proteins show distinct knockout phenotypes for diverse synapses, like RIM-BPs. In small central synapses, the calyx of Held, and inner ear hair cells RIM-BP2 has rather minor effects on neurotransmitter release, by fine-tuning the coupling between $Ca^{2+}$-channels and release sites (*Acuna et al., 2015*; *Grauel et al., 2016*; *Davydova et al., 2014*). Strikingly, we now show that in large MF synapses, the prevailing phenotype of RIM-BP2 loss is an impaired recruitment of Munc13-1 to the active zone, resulting in reduced vesicle docking, priming and consequently neurotransmission. Therefore, RIM-BP2 function depends on the synapse type where it is expressed.

Why does RIM-BP2 functions differ between hippocampal SC and MF synapses? We propose that differences in the structural organization of active zones might be responsible for the observed phenotypes. When comparing apparent distances between active zone protein clusters, clusters of the presynaptic scaffold proteins RIM-BP2 and Munc13-1 were found at larger distances at MF synapses as compared to SC synapses. Also, the distance of the closest neighboring cluster of RIM1 to a given Cav2.1 cluster is larger compared to the distance we previously found at SC synapses (*Grauel et al., 2016*). Therefore, our STED analysis suggests that a single active zone may contain more than one protein cluster of key active zone proteins. Some of these key players might have a differential relative localization between each other at MF synapses in comparison to SC synapses. However, further evidence of sub-active zone organization is required to verify this hypothesis. Moreover, more sophisticated methods of defining protein clusters to belong to the same active zone will be helpful in defining the architecture of the active zone of mammalian central synapses.

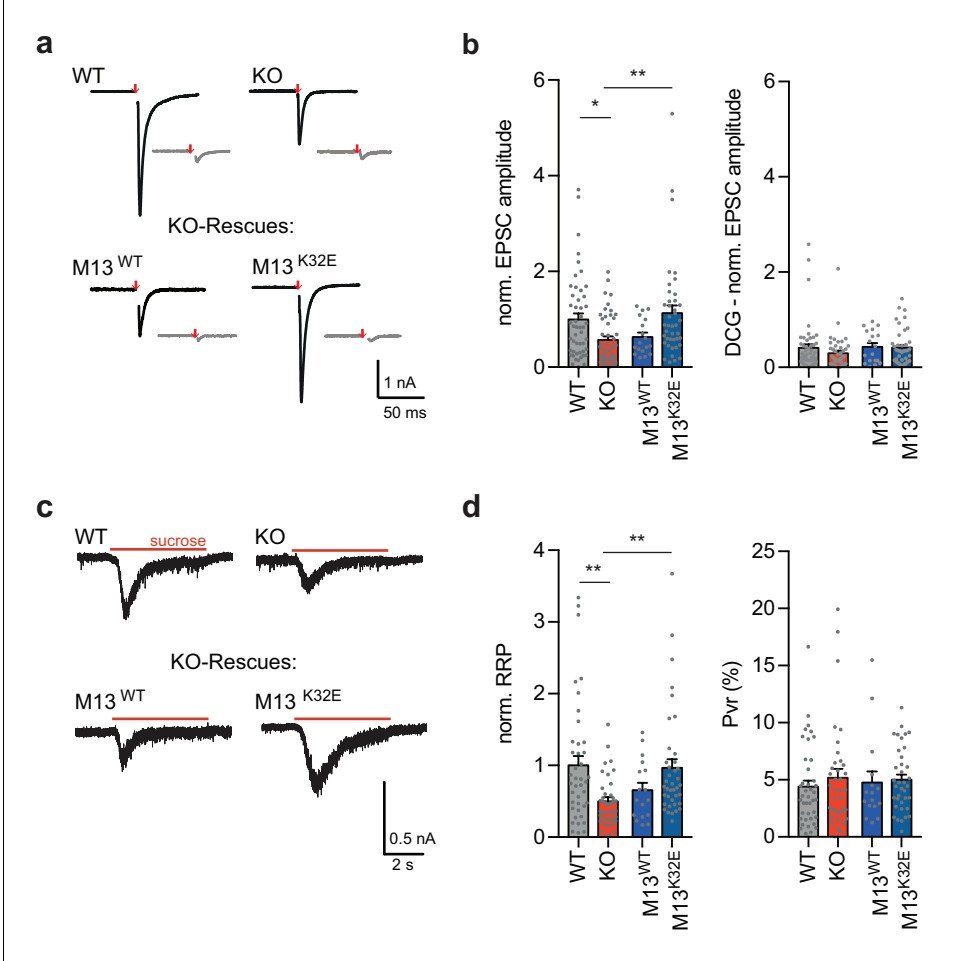

**Figure 6.** Monomeric Munc13-1 rescues vesicle priming in RIM-BP2 KO granule autaptic neurons. (a) Sample traces of evoked EPSC amplitudes before (black) and after DCG IV application (gray) for RIM-BP2 WT and KO neurons and lentiviral-transduced RIM-BP2 KO rescues with either Munc13-1 WT (M13^WT) or Munc13-1 K32E (M13^K32E). (b) Summary graphs of normalized EPSC amplitudes evoked by 2 ms depolarization (red arrow) (EPSC WT (49/3); KO (47/3); M13^WT (18/2); M13^K32E(41/3)) and after DCGIV application. (EPSC-DCG WT (49/3); KO (44/3); M13^WT (18/2); M13^K32E(41/3)) (c) Sample traces and (d) summary graphs of normalized RRP responses elicited by a 5 s application of 500 mM sucrose (RRP WT (41/3); KO (36/3); M13^WT (17/2); M13^K32E(39/3)). Summary graph of the $P_{VR}$ calculated as the ratio of the EPSC charge and the RRP charge (Pvr WT (41/3); KO (38/3); M13^WT (16/2); M13^K32E(37/3)). Values represent mean ± SEM. *p<0.05, **p<0.01. For statistics please see *Figure 6—source data 1*.

DOI: https://doi.org/10.7554/eLife.43243.025

The following source data is available for figure 6:

**Source data 1.** Statistics of data presented in *Figure 6*.

DOI: https://doi.org/10.7554/eLife.43243.026

The exact physiological meaning of active zone protein cluster arrangement is still unresolved but may serve as a correlate to the observed phenotypic differences between WT and KO neurons. However, the distance measurements between clusters of vesicle docking/priming factor Munc13-1 and Ca$^{2+}$-channels we retrieved at MF terminals were in the same range of what was previously reported in the NMJ and CNS of *Drosophila* (*Fulterer et al., 2018*), and were also similar to estimates for Ca$^{2+}$-secretion coupling distances from physiological recordings using efficacy tests of slow Ca$^{2+}$-chelators (*Aravamudan et al., 1999*).

gSTED analysis is superior in defining relative protein localization, but less precise in quantifying absolute protein levels based on fluorescence intensity. In confocal fluorescence measurements performed in parallel to gSTED imaging, we observed that loss of RIM-BP2 led to an overall reduction in Munc13-1 levels at MF synapses. This is in contrast to SC synapses and overall unchanged Munc13-1 levels by western analysis on synaptosomal fraction (*Grauel et al., 2016*). These combined results argue for a specific role of RIM-BP2 in stabilizing Munc13-1 at active zones of MF synapses and consequently the establishment of Munc13-1 clusters. This effect could be indirect *via* RIM, as it is known for SC synapses where vesicle priming is accomplished by the interaction of RIM1 and Munc13-1 independent of RIM-BP2 (*Deng et al., 2011*; *Andrews-Zwilling et al., 2006*), or perhaps through direct interactions *via* RIM-BP2, as previously shown in *Drosophila* (*Böhme et al., 2016*). A critical role of RIM-BP2 in Munc13 recruitment is consistent with the docking/priming phenotype of RIM-BP2 loss in the fly NMJ where the reduction in Munc13 levels is severe upon loss of RIM-BP (*Liu et al., 2011*). The electron micrographs from RIM-BP2 KO MF synapses revealed in addition to a 25% loss of docked vesicles, similarly to what has been previously shown for Munc13-1/−2 deficient central synapses (*Imig et al., 2014*). Putting this together with the gSTED analysis, we speculate that the MF synapse is more sensitive to reductions in Munc13-1 levels, leading to stronger impairment in forming functional Munc13-1 clusters.

Why are Munc13-1 levels differentially affected by RIM-BP2 deletion? One possible explanation is that the expression of interaction partners other than RIM-BP2 varies among synapses. For example, RIM, a critical interaction partner of Munc13-1, is less expressed in the MF synapses compared to SC synapses (*Cembrowski et al., 2016*), and therefore loss of RIM-BP2 cannot be redundantly supported by RIM-Munc13 interactions. In any case, the functional hierarchy of the triple complex formed by RIM-BP2/RIM1/Munc13-1 appears fundamentally different between SC and MF synapses. At MF synapses RIM-BP2 acts first to stabilize RIM1/Munc13-1 at the active zone, whereas at SC synapses, RIM-BP2 impacts synaptic function following RIM/Munc13-1 priming of SV vesicles at the active zone. However, RIM-BP2 dependent stabilization of Munc13-1 still requires RIM activity, since Munc13-1 K32E, the constitutively active form of Munc13-1, but not Munc13-1 WT, is sufficient to rescue the RIM-BP2 MF phenotype (*Figure 6*; and *Deng et al., 2011*; *Andrews-Zwilling et al., 2006*).

Another important question is, if RIM-BP2 alters synaptic release probability at MF terminals. Interestingly, relative $Ca^{2+}$-channel localization at the MF synapses is not affected by the deletion of RIM-BP2, therefore the changes in release probability are rather due to increased coupling distances between $Ca^{2+}$-channels and Munc13-1 as previously shown in *C. elegans* (*Zhou et al., 2013*). Also, the reduction in release probability might be due to the pronounced function of Munc13-2 in RIM-BP2 KO MF synapses (*Rosenmund et al., 2002*), since Munc13-1 clusters but not Munc13-2 clusters abundance and localization are altered upon RIM-BP2 deletion. Still unresolved is the discrepancy between PPR and Pvr measurements in RIM-BP2 deficient MF synapses. Either RRP and Pvr measurements are not correlating at MF granule neurons, or RIM-BP2 specifically alters processes related to PPR but not Pvr, as shown for the synaptic protein Rab3 (*Schlüter et al., 2006*).

In mammalian synapses and *Drosophila* NMJ, RIM-BPs have been shown to biochemically interact with $Ca^{2+}$-channels (*Liu et al., 2011*; *Davydova et al., 2014*), resulting in a severe loss of overall $Ca^{2+}$-channel densities at RIM-BP deficient NMJ's (*Liu et al., 2011*). In contrast, central mammalian synapses show no discernable signs of $Ca^{2+}$-channel dysfunction, but just moderate to weak changes in synaptic transmission, compatible with an impaired $Ca^{2+}$-secretion coupling mechanism (*Grauel et al., 2016*; *Davydova et al., 2014*; *Acuna et al., 2014*). However, the effect is sufficiently strong to cause changes in short term plasticity properties, indicative of a role of RIM-BP2 in regulating release probability and short-term plasticity, similar to what has been observed by loss of the four mammalian rab3 genes (*Schlüter et al., 2006*).

It will be of interest to see whether and through which molecular mechanism loss of RIM-BP2 alters the function of other mammalian synapses in relation to their $Ca^{2+}$-secretion coupling and vesicle priming. While it is clear that factors or structural motifs must exist that determine synaptic diversity such as observed in the described synapses, the composition and spatial organization of such active zone super-organizers remains to be determined.

# Materials and methods

## Key resources table

| Reagent type (species) or resource | Designation | Source or reference | Identifiers | Additional information |
|---|---|---|---|---|
| Genetic reagent (*M. musculus*) | *Rimbp2* | PMID: 27671655 | | Prof. Christian Rosenmund, Prof. Dietmar Schmitz (*Charité*) |
| Recombinant DNA reagent | f(syn)NLS-RFP-P2A-RIMBP2 | This paper | | |
| Recombinant DNA reagent | f(syn)NLS-GFP-P2A-rMunc13.1-Flag | PMID: 28489077 | | |
| Recombinant DNA reagent | f(syn)NLS-GFP P2A-Munc13.1 K32E-Flag | PMID: 28489077 | | |
| Antibody | Bsn (N-terminal) | Abcam | RRID:AB_1860018 | 1:1200 (IHC) |
| Antibody | Cav2.1 (rat aa1921-2212) | Synaptic System | RRID:AB_2619841 | 1:500 (IHC) |
| Antibody | Cav2.1 (rat aa1921-2212) | Synaptic System | RRID:AB_2619842 | 1:500 |
| Antibody | Homer1 (human aa 1–186) | Synaptic System | RRID:AB_10549720 | 1:200 (IHC) |
| Antibody | MUNC13-1 (rat aa 3–317) | Synaptic System | RRID:AB_887734 | 1:150 (IHC) |
| Antibody | MUNC13-2 (rat aa 151–317) | Synaptic System | RRID:AB_2619807 | 1:150 (IHC) |
| Antibody | RIM1 (rat aa 602–723) | BD Pharmigen | RRID:AB_2315284 | 1:200 (IHC) |
| Antibody | RIM-BP2 (rat aa 589–869) | Kind gift of A. Fejtova and Eckart Gundelfinger (Leibniz Institute for Neurobiology, Magdeburg, Germany) | | 1:600/1:1000 (IHC) |
| Antibody | ZnT3 (mouse aa 2–75) | Synaptic System | RRID:AB_2189665 | 1:500 |
| Antibody | Anti mouse AF488 | Invitrogen | RRID:AB_138404 | 1:100/1:200 (gSTED) 1:400 (confocal) |
| Antibody | Anti guinea pig AF594 | Invitrogen | RRID:AB_141930 | 1:100/1:200 (gSTED) 1:400 (confocal) |
| Antibody | Anti rabbit ATTO647N | Active Motif | Cat number #15048 | 1:100 (gSTED) |

## KO mouse generation

RIM-BP2 KO mouse generation, specific deletion of the *Rimbp2* gene, and genotyping was performed as described previously (*Grauel et al., 2016*). All animal experiments were approved by the animal welfare committee of Charité Universitaetsmedizin Berlin and the Landesamt für Gesundheit und Soziales Berlin and carried out under the license (Berlin State Government, T0410/12; T0100/03).

## Slice preparation and electrophysiological recordings

Acute hippocampal slices were prepared as described previously (*Grauel et al., 2016*). In brief, RIM-BP2 KO mice and wild-type littermates of both sexes (4–8 weeks) were anesthetized with Isofluorane and decapitated. The brain was quickly removed and chilled in ice-cold sucrose-artificial cerebrospinal fluid (sACSF) containing (in mM): 50 NaCl, 25 NaHCO3, 10 glucose, 150 sucrose, 2.5 KCl, 1 NaH2PO4, 0.5 CaCl2, and 7 MgCl2. All solutions were saturated with 95% (vol/vol) O2/5% (vol/vol) CO2, pH 7.4.

Slices (300 µm, sagittal or horizontal) were cut with a Leica VT1200S microtome (Wetzlar, Germany) and stored submerged in sACSF for 30 min at 35°C and subsequently stored in ACSF containing (in mM): 119 NaCl, 26 NaHCO3, 10 glucose, 2.5 KCl, 1 NaH2PO4, 2.5 CaCl2 and 1.3 MgCl2 saturated with 95% (vol/vol) $O_2$/5% (vol/vol) $CO_2$, pH 7.4, at RT. Experiments were started 1 to 6 hr after the preparation.

Experiments were conducted in parallel on a comparable number of slices from WT and KO animals prepared at the same experimental day for at least 3 times (biological replicates). Technical replicates were obtained for evoked responses and averaged.

For recordings, slices were placed in a recording chamber continuously superfused with ACSF at RT at a rate of 2.5 ml/min. fEPSPs were evoked by electrical stimulation with patch pipettes filled with ACSF. fEPSPs were recorded with a low-resistance patch-pipette filled with ACSF. Recordings were performed with a MultiClamp 700B amplifier. Signals were filtered at 2 kHz and digitized (BNC-2090; National Instruments Germany GmbH) at 10–20 kHz. IGOR Pro software was used for signal acquisition (WaveMetrics, Inc).

For Mossy fiber recordings, stimulation electrodes were placed in the granule cell layer or in the hilus region. For fEPSP recordings, the recording electrode was placed in stratum lucidum of CA3 region. Mossy fiber origin of recorded signals was verified by frequency facilitation >400% when stimulus frequency was changed from 0.05 to 1 Hz and a complete block of responses upon DCG IV (1 µM; Tocris) application at the end of each experiment. Whole-cell recordings in voltage-clamp mode were performed in CA3 pyramidal cells, with an intracellular solution containing (in mM): K-Gluconate 120; Hepes 10; KCl 10; EGTA 5; MgSO4 2; MgATP 3; NaGTP one and Na-Phosphocreratine 5. Cells were held at −60 mV and series resistance was monitored by delivery of voltage steps prior to each evoked current. Mossy fiber signals were verified by pronounced frequency facilitation and paired pulse facilitation as well as by sensitivity to DCG IV. Paired pulse ratio (50 ms inter-stimulus interval) was measured on the average trace of 20 evoked currents (including possible failures). Failure rate was calculated in 30 traces by detecting the number of traces in which stimulation failed to induce an EPSC. fEPSPs in CA1 were recorded in stratum radiatum after stimulation of the Schaffer collaterals. fEPSPs of associative commissural fibers in area CA3 were recorded in stratum radiatum after stimulation electrodes were places in stratum radiatum of CA3, in the presence of DCG IV (1 µM) to avoid mossy fiber contamination. Amplitudes of EPSCs and fEPSPs were determined by analyzing ±2 ms of the amplitude peak. Data were analyzed with the Igor plug-in NeuroMatic (http://neuromatic.thinkrandom.com/) software. Recordings were only analyzed if the fiber volley remained constant throughout the recording. Statistical analysis was performed with Prism 6 (GraphPad Software).

## Autaptic granule cell culture

Autaptic cultures of Dentate Gyrus Granule cells were prepared as described previously (*Rost et al., 2010*). In brief, the dentate gyrus of P0-P1 RIM-BP2 WT and KO embryos was separated from the hippocampus. After digestion with Papain and trituration, cells were plated on astrocytic microislands (*Arancillo et al., 2013*). Neurons were incubated at 37°C for 14–20 days before the electrophysiological characterization was performed. For rescue experiments, neurons were transduced with lentiviruses 24 hr after plating.

## Lentiviral constructs

Lentiviral constructs used in this study were based on the FUGW vector (*Lois et al., 2002*). The cDNA from mouse RIM-BP2 (NM_001081388) and from rat *Unc-13a* (NM_022861) (*Camacho et al., 2017*) were cloned into an lentiviral shuttle vector after a NLS-GFP-P2A or NLS-GFP-P2A under the control of a human *synapsin-1* promoter. The expression of nuclear RFP or GFP allows to identify transduced neurons. All lentiviruses were provided from the Viral Core Facility of the Charité Berlin and prepared as described before (*Lois et al., 2002*).

## Electrophysiological recordings of autaptic neurons

To pharmacologically identify autaptic granule cells, DCG IV (1 µm) was washed in after each experiment. Only cells where synaptic transmission was inhibited by 70% or more were considered for analysis (*Rost et al., 2010*).

Whole-cell patch-clamp recordings in autaptic neurons were performed as described previously (*Grauel et al., 2016*) at 13–21 days in vitro (DIV) with a Multiclamp 700B amplifier (Molecular Devices). Data were acquired from at least three different cultures (biological replicates). To minimize variability in recordings, about the same number of autapses were recorded from each experimental group each day. Technical replicates were obtained for evoked responses and averaged. Data were normalized to the mean value of the control group of each culture.

The patch pipette solution contained the following (in mM): 136 KCl, 17.8 HEPES, 1 EGTA, 4.6 MgCl$_2$, 4 Na$_2$ATP, 0.3 Na$_2$GTP, and 12 creatine phosphate, and 50 U/ml phosphocreatine kinase (300 mOsm; pH 7.4). The recording chamber was constantly perfused with extracellular solution containing 140 mM NaCl, 2.4 mM KCl, 10 mM Hepes, 2 mM CaCl$_2$, 4 mM MgCl$_2$, and 10 mM glucose (pH adjusted to 7.3 with NaOH, 300 mOsm). Solutions were applied using a fast-flow system. Data were filtered at 3 kHz, digitized at 10 kHz, and recorded with pClamp 10 (Molecular Devices). Data were analyzed offline with Axograph X (AxoGraph Scientific) and Prism 6.

EPSCs were evoked by a 2 ms depolarization to 0 mV from a holding potential of −70 mV. PPRs were calculated as the ratio from the second and first EPSC amplitudes with an interstimulus interval of 25 ms. The RRP size was calculated by integrating the transient current component of 5 s evoked by application of extracellular hypertonic 500 mM sucrose solution. Miniature EPSC (mEPSC) amplitude and frequency were detected using a template-based algorithm in Axograph X.

## Immunohistochemistry, time gated STED microscopy and image analysis

Immunohistochemistry was performed as described previously (*Grauel et al., 2016*). Notably, the antibody against RIM1 (RRID: AB_2315284, BD Pharmigen) used in this study is not commercially available anymore. To repeat these results, other RIM1 antibodies need to be validated beforehand. Conventional confocal tile scans of RIM-BP2 immunofluorescence in the hippocampus were acquired with a Leica SP8 laser confocal microscope equipped with a 20 × 0.7 N.A. oil immersion objective.

Following immunostaining, sagittal cryosections (10 μm) of RIM-BP2 WT and KO brains were imaged by gSTED with a Leica SP8 gSTED microscope (Leica Microsystems) as described previously (*Grauel et al., 2016*). Within each independent experiment, RIM-BP2 KO and WT samples were imaged with equal settings. Single optical slices were acquired with an HC PL APO CS2 100×/1.40 N.A. oil objective (Leica Microsystems), a scanning format of 1,024 × 1,024, eight bit sampling, and 4.5 zoom, yielding a pixel dimension of 25.25 nm and 25.25 nm in the x and y dimensions, respectively. Four to eight super-resolved images were acquired per a single brain section of each mouse analyzed (n indicates the number of mice analyzed per each data set). MF were imaged in the CA3 stratum lucidum close to CA3 pyramidal cell bodies, where MF boutons make contact on proximal dendritic spines of CA3 pyramidal neurons. CA3-CA1 synapses were acquired in the CA1 stratum radiatum. Raw dual- and triple-channel gSTED images were deconvolved with Huygens Professional software (Scientific Volume Imaging) using a theoretical point spread function automatically computed based on pulsed- or continuous-wave STED optimized function and the specific microscope parameters. Default deconvolution settings were applied.

Experiments were performed at least two times on different mice (biological replicates).

For cluster distance analysis, deconvolved images were thresholded and segmented by watershed transform with Amira software (Visualization Sciences Group) to identify individual clusters and to obtain their x and y coordinates. Within the same independent experiment, the same threshold and segmentation parameters were used. According to the lateral resolution achieved, clusters with a size smaller than 0.0025 μm$^2$ were not considered for analysis. To select Munc13-1 and Ca$_V$2.1 clusters within the ZnT3 +area, a mask was created applying a threshold on deconvolved ZnT3 +confocal images with Amira software (Visualization Sciences Group). Hundreds to thousands of clusters per single image were automatically analyzed. The average number of clusters at specific distances and the *k*-nearest neighbor distance were analyzed with a MATLAB custom-written script, as previously described (*Grauel et al., 2016*). More in details, in the first step the script determined the Euclidean distance between all possible cluster pairs in two channels in a matrix. The number of clusters in channel 1 found within 50 nm, 75 nm, 100 nm, 125 nm, 150 nm, 200 nm and 300 nm distances from each single cluster of channel two was calculated and averaged for all particles found in channel 2. To precisely identify at which specific distance changes in clustering may occur, the mean number of channel one clusters found at sampling distances from channel two was expressed in

distance intervals (0–50 nm, 50–75 nm, 75–100 nm, 100–125 nm, 0.125–200 nm and 200–300 nm). *K*-nearest neighbor distance analysis was similarly based on the matrix containing the distances between all particles in both channels: the distances of all particles in channel one to the ones in channel two were sorted in ascending order to find the *k*-nearest neighbor. *K* was set to 1, 2, 3, 4 and 5. The *k* distance values were then averaged on the number of clusters in channel 2. Values retrieved for each image were then averaged per mouse. Data from independent experiments were pooled.

Line profile measurements of distances between clusters was performed in Imagej (NIH) on the same deconvolved gSTED images used for the semi-automated analysis. Well defined side view or planar synapses within areas with high density of active zones corresponding to MF synapses were manually traced with the line profile tool (thickness 10 pixels/~250 nm), using the ImageJ Macro (Macro_plot_lineprofile_multicolor from Kees Straatman, University of Leicester, Leicester, UK). Intensity values from individual synapses were exported to Excel. Up to six line profiles per image were consider for analysis. Local maxima were calculated with the SciPy 'argrelmax' function in order to obtain peak intensities for different image channels and peak-to-peak distances. Values were then averaged per mouse. Code has been made available via Github: (*Gimber, 2019*; copy archived at https://github.com/elifesciences-publications/LineProfileAnalysisWorkflow).

Measurements of Munc13-1 intensity were performed in ImageJ (version 1.51 w) on confocal images acquired in parallel to the gSTED images. Mean pixel gray values were measured for each whole image (25.83 µm x 25.83 µm). Four to eight confocal images per mouse were quantified and values averaged per mouse. Data from independent staining were pooled and analyzed.

## Electron microscopy

Acute Hippocampal slices (150 µm) were prepared as described above and frozen at RT using an HPM 100 (Leica) supported with extracellular solution containing 15% Ficoll.

Slices from at least three different WT and KO animals were frozen and processed in parallel (biological replicate). After freezing, samples were transferred into cryovials containing 1% glutaraldehyde, 2% osmium tetroxide, and 1% ddH$_2$O in anhydrous acetone in an AFS2 (Leica) with the following temperature program: −90℃ for 72 hr, heating to −60℃ in 20 hr, −60℃ for 8 hr, heating to −30℃ in 15 hr, −30℃ for 8 hr, heating to −20℃ in 8 hr. After staining with 1% uranyl acetate, samples were infiltrated and embedded into Epon and backed 48 hr at 60℃. Serial 40 nm sections were cut using a microtome (Leica) and collected on formvar-coated single-slot grids (Science Services GmbH). Before imaging, sections were contrasted with 2.5% (wt/vol) uranyl acetate and lead citrate. Samples were imaged in a FEI Tecnai G20 TEM operating at 80–120 keV and images taken with a Veleta 2K x K CCD camera (Olympus) and analyzed with a custom-written ImageJ (NIH) and MATLAB (The MathWorks, Inc) script. The size of the post-synaptic density is defined as the length of the prominent electron dense material at the post-synaptic side of the synapse. Small clear vesicles were defined by their diameter between 30–55 nm, whereas large clear vesicles were defined >60 nm.

## Statistical analysis

For electrophysiological experiments in brain slices, numbers of experiments are indicated in n/N, while n represents the number of brain slices and N the number of animals. Sample size estimation was done as published previously (*Breustedt et al., 2010*).

For confocal and gSTED analysis, statistical analysis was done with Prism (Graphpad) and SPSS Statistics software (IBM), respectively. Normality was assessed checking histograms and Q-Q plots. Pairwise comparisons were analyzed with the Mann–Whitney *U* test. Significance threshold α was set to 0.05. Only p values less than 0.05 were considered significant. Values corresponding to one WT animal measured displaying an extreme outlier were excluded from the whole MF Munc13-1/Ca$_V$2.1 data analysis, based on SPSS estimation of extreme values (value >Q3+3*IQR). This WT mouse was excluded also from Munc13-1 intensity measurements at MF terminals. Values are expressed as mean ± SEM, and *n* indicates the number of animals tested. Sample size estimation was done as published previously (*Grauel et al., 2016*).

For autaptic electrophysiological experiments, statistical analysis was done in Prism (Graphpad). First, the D'Agostino-Pearson test was applied to check whether data are normally distributed. If

data were normally distributed, statistical significance was determined by using one-way ANOVA followed by Turkey *post hoc* test. For data which were not normally distributed, statistical significance was assessed using non-parametric Kruskal-Wallistest with Dunn's *post hoc* test. Values are expressed as mean ± SEM, and *n* indicates the number of recorded autapses. Sample size estimation was done as published previously (*Camacho et al., 2017*; *Liu et al., 2016*).

For electron microscopy experiments, the D'Agostino–Pearson omnibus test was used to check for normal distribution of data. For WT vs. KO comparison, an unpaired *t* test with Welch's correction was used for normally distributed data and the Mann–Whitney *U* test was used for not normally distributed data. Values are expressed as mean ± SEM, and *n* indicates the number of active zones analyzed. Sample size estimation was done as published previously (*Grauel et al., 2016*).

## Acknowledgements

This work was supported by the Deutsche Forschungsgemeinschaft (Collaborative Research Grant SFB 958 [to DS (A5), SJS (A3, A6), CR (A5)] and Excellence Strategy – EXC-2049–390688087 (to DS, SJS, and CR). We thank Melissa Herman and Gülcin Vardar for providing critical advice for writing the manuscript and Alexander Walter for providing the gSTED analysis script and comments. We thank Berit Söhl-Kielczynski, Anke Schönherr, Susanne Rieckmann und Lisa Züchner for excellent technical support; and Jörg Breustedt for discussions. We thank the Charité viral core facility for virus production and the cellular imaging facility of the Leibniz-Forschungsinstitut für Molekulare Pharmakologie (FMP) for the use of the gSTED microscope. We thank Anna Fejtova and Eckart Gundelfinger for the RIM-BP2 antibody.

## Additional information

### Funding

| Funder | Grant reference number | Author |
| --- | --- | --- |
| Deutsche Forschungsgemeinschaft | SFB 958 A5 | Christian Rosenmund Dietmar Schmitz |
| Deutsche Forschungsgemeinschaft | SFB 958 A3 | Stephan J Sigrist |
| Deutsche Forschungsgemeinschaft | SFB 958 A6 | Stephan J Sigrist |
| Deutsche Forschungsgemeinschaft | EXC-2049–390688087 | Dietmar Schmitz Stephan J Sigrist Christian Rosenmund |

The funders had no role in study design, data collection and interpretation, or the decision to submit the work for publication.

### Author contributions

Marisa M Brockmann, Conceptualization, Formal analysis, Supervision, Funding acquisition, Investigation, Visualization, Writing—original draft, Project administration, Writing—review and editing; Marta Maglione, Conceptualization, Formal analysis, Investigation, Visualization, Writing—original draft, Writing—review and editing; Claudia G Willmes, Formal analysis, Investigation; Alexander Stumpf, Investigation, Writing—review and editing; Boris A Bouazza, Laura M Velasquez, M Katharina Grauel, Prateep Beed, Investigation; Martin Lehmann, Software; Niclas Gimber, Jan Schmoranzer, Formal analysis; Stephan J Sigrist, Conceptualization, Supervision, Funding acquisition, Investigation, Writing—review and editing; Christian Rosenmund, Conceptualization, Supervision, Funding acquisition, Project administration, Writing—review and editing; Dietmar Schmitz, Conceptualization, Supervision, Funding acquisition, Writing—review and editing

### Author ORCIDs

Marisa M Brockmann (iD) https://orcid.org/0000-0002-1386-5359
M Katharina Grauel (iD) https://orcid.org/0000-0003-3542-0606

Stephan J Sigrist (iD) https://orcid.org/0000-0002-1698-5815
Christian Rosenmund (iD) https://orcid.org/0000-0002-3905-2444

## Ethics

Animal experimentation: All animal experiments were approved by the animal welfare committee of Charité Universitaetsmedizin Berlin and the Landesamt für Gesundheit und Soziales Berlin and carried out under the license (Berlin State Government, T0410/12; T0100/03).

## Decision letter and Author response

Decision letter https://doi.org/10.7554/eLife.43243.029
Author response https://doi.org/10.7554/eLife.43243.030

## Additional files

### Supplementary files

• Transparent reporting form
DOI: https://doi.org/10.7554/eLife.43243.027

### Data availability

All data generated are shown as scatter plots in each graph. Code has been made available via Github (https://github.com/ngimber/LineProfileAnalysisWorkflow/releases/tag/1.0.0; copy archived at https://github.com/elifesciences-publications/LineProfileAnalysisWorkflow).

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
