## [Decision Letter]

Thank you for submitting your article "RIM-BP2 primes synaptic vesicles via recruitment of Munc13-1 at hippocampal mossy fiber synapses" for consideration by *eLife*. Your article has been reviewed by Gary Westbrook as the Senior Editor, a Reviewing Editor, and three reviewers. The following individual involved in the review of your submission has agreed to reveal his identity: Craig Garner (Reviewer #2). The reviewers have discussed the reviews with one another and the Reviewing Editor has drafted this decision to help you prepare a revised submission.

Summary:

The project investigates the functional role of the presynaptic scaffold protein RIM-BP2 at three hippocampal synapses. The authors report that deletion of RIM-BP2 mossy fiber (MF) synapses strongly impairs neurotransmission. In contrast, input-output function at other excitatory hippocampal synapses was not impaired. The data are confirmed in cultured neurons forming autapses, where it is further shown that lentiviral expression of Munc13-1KE bypasses the need for RIM-BP2. Finally, the authors provide STED analyses in slices, where they assess protein cluster localization by applying the 'nearest neighbor analyses' (NN), and find changes in Munc13-1 clusters, but not in Munc13-2, Ca channels and RIM clusters. Although the reviewers judged the observation that RIM-BP2 has important roles in synaptic release specifically at MF terminals as interesting, they had several major concerns, which will be summarized in the following:

1) In general, the reviewers found the description of many experiments very superficial, lacking sufficient information for comprehending the way the conclusions were reached. The illustrations of the findings are also suboptimal. (a) For example, the authors did not even mention the experiments described in Figure 6C,D in the Results section. (b) In Figure 4 the EM images illustrating the synapses are very small. The authors should present low power images to provide an overall view of the tissue, and provide multiple high power images of many synapses. (c) Why did the authors use high-pressure cryofixation? Please explain. (d) The authors quantitatively analyze the distribution of RIM, Cav2.1 and homer clusters. The description of these experiments is extremely superficial in the Results section. (e) The reader could not figure out how many synapses from how many slices and from how many animals the authors analyzed. (f) There is a lack of information in the results or the legends about the way the authors clustered their data. Finally, the manuscript contains many typos and language imprecisions and therefore requires very carefully edited.

2) Two reviewers formulated significant concerns about the STED analysis, which needs very careful revision. The reviewers wish to see data on how many clusters they find, what method was used for determining the clusters, how the distances between clusters has been measured and more importantly how the authors distinguished between clusters within an active zone and between active zones. The manuscript must be improved in this respect. Detailed comments see below in A-C.

A) The NN-STED analysis is not very meaningful at MF terminals. While NN is insightful for methods that localize individual molecules, it is not for STED, which does not have this capacity. It is uncertain whether any protein cluster is within MF terminals or may be in other synapses around. The nature of biochemical interactions of these proteins suggests that they are intermixed as they all bind to one another, it is not clear that there are separable RIM or CaV2 or Munc13 clusters at distances of 200 nm of each other within an active zone. It is also not clear whether clusters of different proteins belong to the same active zone or even the same synapse. Some of these points are reflected in the data. In Figure 2, it is unclear how RIM, on average, ends up being closer to Homer1 than to CaVs, if the three belong to the same synapse (CaVs sit in the membrane that separates RIM and Homer). Figure 3 claims a reduction in Munc13 levels, but what the author's measure in their analysis is cluster numbers, not levels. It would be much preferable to have a measurement of Munc13 levels within a bouton or within an active zone. This could easily be done in experiments in cultured Dentate Gyrus neuron autapses. The authors should provide either the information on cluster analyzes or find a better way to analyze their STED images.

B) On similar lines: One fundamental issue relating the overall conclusion is the number of clusters of each molecules and their spatial separations within the AZ. Given an average AZ area of 0.06 um2, the diameter of such average AZ (assuming disc shape) would be ~280 nm. The authors do not tell the number of clusters for each proteins, but their separation distances are around 260 and 110 nm. The reviewer would like to see the number of clusters per AZ for each presynaptic AZ molecules and their spatial arrangements that is compatible with the measured inter-cluster distances. Without knowing the numbers, the reviewer cannot comprehend whether such cluster separation distances can be consistent with the size of the AZ or not.

C) How were distances between 'clusters' measured? For example, bassoon seems to form a homogeneous line in Supplementary Figure 1A and RIM-BP and Munc13-1 form some clusters. The authors should demonstrate on the figure how they measured these distances and they should provide more images from which the readers could have an impression of how similar or dissimilar the synapses were with respect to these proteins.

Related to the problem of cluster distance quantification: The authors performed two sets of colocalization experiments: Cav2.1 vs. RIM1 and Munc13-1 vs. RIM BP2. The distances between these clusters were ~ 250 and 170 nm. What is the spatial relationships among them?

3) Another major shortcoming of the manuscript is the lack of detailed demonstration of how RIM-BP2 affects Schaffer collateral (SC) synapses. First, the authors did not perform all localization experiments in SC synapses in parallel with that of mossy fiber synapses. In addition, no functional experiment is presented that would test the prediction of the altered molecular composition of the SC synapses. Please provide the comparative functional data.

4) A further important addition is to do a better paired-pulse ratio (PPR) analysis in cultured dentate gyrus neurons to assess probability of vesicular release (Pvr). The increased PPR strongly suggest a decrease in Pvr, but the analysis is limited to a single inter-spike-interval. The authors use the sucrose method as a measure of Pvr and argue that Pvr is unchanged, but it remains unclear whether this is a valid way to quantify Pvr at cultured dentate gyrus synapses. A complete analysis of PPRs at cultured dentate gyrus neurons and in hippocampal slices should be performed at various inter-stimulus intervals. This would better connect the slice and culture data sets, and could either support decreased Pvr, or reject this hypothesis. If PPRs indicate decreased Pvr, then this should be better stated in the Title and the Abstract.

Full reviews included for reference:

*Reviewer #1:*

This study investigates the functional role of the presynaptic scaffold RIM-BP2 at three hippocampal synapses. The authors report that deletion of RIM-BP2 strongly impairs mossy fiber neurotransmission, but input-output function at other excitatory hippocampal synapses is not impaired. These phenotypes are confirmed in cultured autaptic neurons, where it is further shown that lentiviral expression of Munc13-1KE bypasses the need for RIM-BP2. Finally, the authors provide STED analyses in slices, where they assess protein cluster localization with "nearest neighbor analyses" (NN), and find changes in Munc13-1 clusters, but not Munc13-2, Ca channels and RIM.

The observation that RIM-BP2 has important roles in release specifically at mossy fiber synapses is interesting, relevant, and established with the represented work. I strongly support publishing this finding in *eLife*. However, the paper has several problems that should be addressed first. While only a limited number of experiments are needed, particularly the analyses of the STED data is problematic. While a link to Munc13 appears likely, this needs to be better addressed, Abstract and Title should express that this is likely indirect, and it should be better discussed upfront that the phenotype probably arises from a combination of reduced RRP and reduced pvr.

1) The NN STED analysis is not very meaningful at MF terminals. While NN is insightful for methods that localize individual molecules (storm, palm or immuno-gold EM), it is not for STED which does not have this capacity. It is uncertain whether any protein cluster is within MF terminals or may be in other synapses around. More, the nature of biochemical interactions of these proteins suggests that they are intermixed as they all bind to one another, it is not clear that there are separable RIM or CaV2 or Munc13 clusters at distances of 200 nm of each other within an active zone. It is also not clear whether clusters of different proteins belong the same active zone or even the same synapse. Some of these points are reflected in the data. In Figure 2, it is unclear how RIM, on average, ends up being closer to Homer1 than to CaVs if the three belong to the same synapse (CaVs sit in the membrane that separates RIM and Homer). Figure 3 claims a reduction in Munc13 levels, but what the authors measure in their analysis is cluster numbers, not levels. In the end, it would be much preferable to have a measurement of Munc13 levels within a bouton or within an active zone. This could easily be done in experiments in cultured dg neuron autapses. The authors should also find a better way to analyze their STED images, or remove the current analysis.

2) Most aspects of the phenotype look like a "reduced RIM-phenotype" in the RIM-BP mutants: A reduction in RRP, loss of docked and tethered vesicles (Munc13 knockouts only have a loss of vesicles within 2 nm and increased vesicles within 5-30 nm), and the phenotype can be bypassed with KE Munc13 (which mimics monomeric Munc13 that is independent of RIM) but not regular Munc13 (which requires RIM for activation). These data strongly predict reduced RIM levels or aberrant RIM localization in MF terminals upon RIM-BP2 KO, the only experiment that speaks against this the STED nearest neighbor analysis, but see 1. RIM localization and levels should be better assessed, the most straightforward would be to do this in dg autapses.

3) The increased paired pulse ratios strongly suggest a decrease in pvr (but the analysis is limited to a single ISI and hence not very strong). The authors use the division of the EPSC by the sucrose EPSC as a measure of pvr and argue that pvr is unchanged, but I could not find literature to establish that it is valid to quantitatively determine pvr at cultured dg synapses using this indirect method. Reduced pvr in addition to the RRP deficit makes sense in terms of everything that is known about RIM-BP and its roles, its interactions with RIM, and the observed PPR changes in Figure 5. A complete analysis of PPRs at cultured dg neurons and in hippocampal slices should be performed at various interstimulus intervals. This would better connect the slice and culture data sets, and could either support decreased pvr, or reject this hypothesis. If PPRs indicate decreased pvr, this should be better stated in Title and Abstract, there is no need for RIM-BP to have an isolated effect on RRP for this study to be interesting.

*Reviewer #2:*

This manuscript describes experiments to compare the contribution of RIM-BP2 to synaptic transmission at DG-CA3 mossy fiber (MF) boutons to small CA3-CA1 excitatory synapses. Electrophysiological and morphological studies reveal that RIM-BP2 loss of function only modestly affect CA1 synaptic transmission and Ca^2+^ secretion coupling, while its loss at MF boutnd has a strong impact on neurotransmitter release by promoting SV docking and priming via Munc13-1. The data collected strongly supports the claims of the authors, namely that the nano-environment and complexes formed by RIM-BP2 with the active zones of these two synapse, imparts function differences. Although molecular diversity at synapses has been long thought to be a possible mechanism for contributing to functional difference at synapses, there are few examples where this has been demonstrated to be the case. This study thus provides compelling evidence that RIM-BP2 is one such regulator.

Over all the data presented are of very high quality. A particular strength is the combination of data from acute slices, where the identity of each synapses is well established, and autaptic cultures with the electrophysiological properties of neurotransmitter release can be best evaluated. Also strong are the EM and superresolution data, providing information on changes in molecular distances of different active zone proteins and docking of SVs.

One concern requiring some attention is the author's use of the term levels of synaptic protein. (e.g. Results section; Discussion section). It is not self-evident that any measures of levels were perform. Cluster # yes, but no intensity values. This should be amended. Also, any statement on changes in levels should Western data in addition to (Discussion section) RNAseq data, which would reflect changes in expression levels of these key proteins.

*Reviewer #3:*

The present MS describes physiology and anatomy experiments, addressing the role of RIM-BP2 in neurotransmission. The authors demonstrate that the loss of RIM-BP2 has a differential effect in hippocampal MF vs. SC synapses, suggesting that the same protein could serve different roles in distinct central synapses. The combined superresolution gSTED microscopy and electrophysiology approach is powerful in revealing potential molecular mechanisms underlying distinct function. In general, the reviewer finds the description of many experiments very superficial, lacking sufficient information for comprehending the way the conclusion was reached. The illustrations of the findings are also suboptimal.

One fundamental issue relating the overall conclusion is the number of clusters of each molecules and their spatial separations within the AZ. Given an average AZ area of 0.06 um2, the diameter of such average AZ (assuming disc shape) would be ~280 nm. The authors do not tell the number of clusters for each proteins, but their separation distances are around 260 and 110 nm. The reviewer would like to see the number of clusters per AZ for each presynaptic AZ molecules and their spatial arrangements that is compatible with the measured inter-cluster distances. Without knowing the numbers, the reviewer cannot comprehend whether such cluster separation distances can be consistent with the size of the AZ or not.

It seems that the authors are not aware of the fact that hippocampal MF AZs contact distinct postsynaptic target cell types with very different functional properties. It was surprising to read that the authors lumped together all MF AZs irrespective of their postsynaptic target cells.

Another major shortcoming of the MS is the lack of detailed demonstration of how RIM-BP2 affects SC synapses. First, the authors did not perform all localization experiments in SC synapses in parallel with that of MF synapses. In addition, no functional experiment is presented that would test the prediction of the altered molecular composition of the SC synapses.

The authors analyse the distribution of RIM, Cav2.1 and homer clusters quantitatively. The description of these experiments is extremely superficial in the Results section. The reader cannot figure out how many synapses from how many slices and from how many animals the authors analysed. Nothing is written in the results or the legends about the way they clustered their data. Similar, the reader has no idea about the way the authors measured distances between such 'clusters'. For example, bassoon seems to form a homogeneous line in Supplementary Figure 1A and RIM-BP and Munc13-1 form some clusters. The authors should demonstrate on the figure how they measured these distances and should also provide more images from which the readers could have an impression of how similar or dissimilar the synapses were with respect to these proteins.

The reviewer has problems understanding the beautiful LM gSTED images. For example, in Figure 3G, the entire area seems to be tiled with green and red areas. How can the authors determine the boundaries of synapses? Where does one end and the second starts?

In subsection “RIM-BP2 docks synaptic vesicles *via* the specific recruitment of Munc13-1 at MF Synapses”, the authors describe a large change in the distance between Cav2.1 and Munc13-1 clusters. From the image in Figure 3A, the most dramatic effect of RIM-BP KO is the reduction of Munc13-1 protein. The functional interpretation of these two effects is completely different. The first would indicate a much reduced Pv whereas the second a dramatic reduction in N (number of docking sites).

Many parts of the experiments are superficially described in the manuscript. For example, the authors did not even mention the experiments described in Figure 6C,D in the Results section.

The authors describe the quantification of their gSTED data on bassoon, Munc13-1 and RIM-BP2 in Supplementary Figure 1. The illustration of that data together with those obtained in KO mice should be moved to the main figures.

The authors performed two sets of colocalization experiments: Cav2.1 vs. RIM1 and Munc13-1 vs. RIM BP2. The distances between these clusters were around 250 and 170 nms. What is the spatial relationships among them (see my major point as well)?

Figure 2B, Figure 3B,C,H,I: The authors should not present their data as% of control, but should show the number of clusters they detect.

Figure 4: The EM images illustrating the synapses are very small. The authors should present low power images to provide an overall view of the tissue and also have multiple high power images of many synapses.

Why did the authors use high-pressure cryofixation?

[Editors' note: further revisions were requested prior to acceptance, as described below.]

Thank you for resubmitting your work entitled "RIM-BP2 primes synaptic vesicles via recruitment of Munc13-1 at hippocampal mossy fiber synapses" for further consideration at *eLife*. Your revised article has been favorably evaluated by Gary Westbrook (Senior Editor), a Reviewing Editor, and three reviewers. The manuscript has been improved but there are a few remaining issues that we would like you to address before acceptance. For your information, we have also included the comments of the reviewers prior to the reviewer/editor discussion.

Summary:

After careful re-reading of the revised manuscript, all reviewers acknowledge the revision by the authors, the importance of the study and the fact that the study should be published at *eLife*. However, all reviewers agreed that the details of the image analysis and cluster analysis could be improved. The limitation being that it is unclear whether a cluster belongs to the same active zone or not. Therefore, please further discuss the details of the cluster analysis. In this context, please tone down the importance of the cluster analysis in the Abstract. Moreover, emphasize the problem with the availability of the RIM antibody that readers understand the limitations.

*Reviewer #1:*

I continue to strongly support this paper and it is fine to accept it at *ELife* as is. The observation that RIM-BP2 knockout has strong effects on MF but not other hippocampal excitatory synapses, and that this role is mediated through Munc13-1 is interesting and important. There are a few points, however, the authors may want to consider:

a) I continue to think that there is too much emphasis on the nearest neighbor analysis. The authors now do a good job in describing some of its limitations, but I am still not convinced that the fundamental assumption that underlies this analysis – that there are separable clusters of RIM-BP, CaVs, Munc13s, etc. within an active zone – makes sense, at least not for all of this proteins. It is fine for me to remain in disagreement on this point. The paper has a very important message regardless.

b) Subsection “RIM-BP2 docks synaptic vesicles *via* the specific recruitment of Munc13-1 at MF synapses”: the docking deficit is ~25%, not 50%, and the text refers to the wrong figure panel.

*Reviewer #2:*

In the revised manuscript the authors have thoroughly addressed each of the concerns raised by the previous reviewers. In its currently form the data presented in the manuscript strongly support the author's conclusions. In brief, they have examined whether the molecular diversity of active zones could contribute to difference in synaptic function. Here they focused on the contribution of RIM-BP2 at hippocampal mossy fiber and small CA1 synapses. While RIM-BP2 loss of function has only modest effects on CA1 synapses, it loss at MF boutons results in quite profound changes in synaptic transmission. Their light, EM, super-resolution and functional analysis point to prominent role for RIM-BP2 in positioning Munc13-1 at MF synapses but not at CA1 synapses. Importantly, associated defects at MF-synapse can be rescued by a constitutively active Munc13 (K32E) indicating that RIM-BP2 faciliates the localization and activation of Munc13-1 at these synapses.

Overall this is a very nice manuscript that reads well and support the major claims of the authors. I would support the publication of this manuscript in it current form, without the addition of new data.

*Reviewer #3:*

In their rebuttal, the authors pulled together the comments of the three reviewers in an unusual way and aimed to address the concerns together. This hampered to comprehension of the responses a bit and also lead to some confusions! It seems that even the authors were confused by the way they addressed the concerns!

Intra- or inter-release site clustering: The authors admit that they do not really know which clusters are within an AZ and which ones are in different AZs. They argue that because the mean AZ diameter is 390 nm therefore two clusters that are within this distance must belong to one AZ and the ones with larger separation are in separate AZs. This is an incorrect and way oversimplified argument. First, without knowing the inter AZ distance distribution, such statement cannot be made. If the shortest inter-AZ distance is only 250 nm, then an inter-cluster distance of 300 nm might still reflect an inter-AZ arrangement. Second, as the authors pointed out this 390 nm is the mean AZ diameter. It is well known that there is tremendous variability in the AZ diameters, thus even a 500 nm inter-cluster distance could still represent a within-AZ arrangement in large AZs.

Although the authors argue that now they can determine the number of clusters per AZ for many proteins, the reviewer is not convinced. One is the consequence of the uncertainty of delineating AZs (see above) and the second is the lack of convincing demonstration of the objective clustering.

The authors misinterpreted the reviewer's concern regarding the distinct postsynaptic targets of MFs. Indeed, the reviewer referred to the fact that granule cell axons contact CA3 PNs as well as a large variety of GABAergic INs. The important point is that the functional and structural properties of these synapses are highly heterogeneous. Thus, the population comparisons of the function (field EPSP) to random gSTED images is fundamentally flawed. Meaningful comparisons would require the molecular identification of every postsynaptic target cell type in the STED experiments and the intracellular recordings all possible postsynaptic cell types (CA3 PNs and every type of INs).

---

## [Author Response]

[…] Although the reviewers judged the observation that RIM-BP2 has important roles in synaptic release specifically at MF terminals as interesting, they had several major concerns, which will be summarized in the following:1) In general, the reviewers found the description of many experiments very superficial, lacking sufficient information for comprehending the way the conclusions were reached. The illustrations of the findings are also suboptimal. (a) For example, the authors did not even mention the experiments described in Figure 6C,D in the Results section. (b) In Figure 4 the EM images illustrating the synapses are very small. The authors should present low power images to provide an overall view of the tissue, and provide multiple high power images of many synapses. (c) Why did the authors use high-pressure cryofixation? Please explain. (d) The authors quantitatively analyze the distribution of RIM, Cav2.1 and homer clusters. The description of these experiments is extremely superficial in the Results section. (e) The reader could not figure out how many synapses from how many slices and from how many animals the authors analyzed. (f) There is a lack of information in the results or the legends about the way the authors clustered their data. Finally, the manuscript contains many typos and language imprecisions and therefore requires very carefully edited.

We thank the reviewers for raising these points. We restructured the text and added more information to describe each conducted experiment in more details.

a) We are sorry to have omitted this point. The description of the graph is now added to the current manuscript.

b) For a better illustration of the EM results, we added more pictures for MF synapses with low and high resolution (Figure 4).

c) In the present study, we choose to perform our EM analysis of docked synaptic vesicles using highpressure freezing with subsequent chemical freeze substitution. High pressure freezing emerged in recent years as a new standard in the analysis of synaptic structures, and in particular in description of processes such as synaptic vesicle docking. The main reasons are that this approach appears to have less experimental artifacts. See for example effects of deletion of Munc13 leads in chemical fixation to no changes in vesicle docking, while it does so in high pressure frozen samples (Camacho et al., 2017; Varoqueaux et al., 2002). As structures are arrested on a ms time scale movement, diffusion-based artifacts are reduced. Second, chemical fixation dehydrates the fixed tissue, induces more sample shrinkage and triggers synaptic vesicle fusion due to hyperosmolarity of fixation solutions.

d) We agree with the reviewers and included a more detailed description of these experiments in the main text and in the figure legend, inclusive of a new data analysis (Figure 2D-H).

e and f) In the first version of our manuscript, we stated the data processing and number of conducted experiments exclusively in Supplementary file 1, in the statistics and in the method part. We now added the number of biological replicates per group to the figure legends, to provide better accessibility to this information. We also included more information regarding our experiments and analysis in the Material and methods section.

2) Two reviewers formulated significant concerns about the STED analysis, which needs very careful revision. The reviewers wish to see data on how many clusters they find, what method was used for determining the clusters, how the distances between clusters has been measured and more importantly how the authors distinguished between clusters within an active zone and between active zones. The manuscript must be improved in this respect. Detailed comments see below in A-C.

We thank the reviewers for raising these important points and appreciate the possibility to explain our strategy point by point:

1) Definition of protein clusters and determination of cluster distances:

We identified protein clusters in deconvolved 2D gSTED images based on an uneven fluorescence signal distribution. We used thresholding of single channel images, followed by watershed segmentation to define AZ protein clusters. Per image, several hundreds to thousands of clusters were analyzed. The x and y coordinates of the center of mass for each cluster were retrieved by Amira software and Euclidean distances between clusters formed by two AZ proteins of interest were determined with a custom-written MATLAB script, previously published in Grauel et al., 2016. Values retrieved for each image were then averaged per mouse. We now show graphs indicating the average number of protein clusters found per each animal analyzed (Figure 2E, Figure 3C,D,I,J, Figure S5C,D,H,J).

Our main goal was to determine if loss of RIM-BP2 impacts the AZ localization of the relevant proteins at different synapse types, looking therefore for relative changes.

2) Differentiating intra-AZ from inter-AZ clusters/Estimating intra-AZ distances for AZ protein clusters:

As correctly pointed out by the reviewers, our gSTED analysis using nearest neighbor detection does a priori not allow us to discriminate between clusters belonging to a single AZ or to nearby AZs. However, the average dimensions of individual AZs have been systematically measured at MF boutons using electron microscopy Rollenhagen et al., 2007. From these data the average AZ diameter is near 390 nm. We therefore first rank in our nearest neighbor analysis the neighbors according to their proximity, and subsequently define protein clusters with an inter-clusters distance d*_k_*<390 nm to be putative within individual AZs. This leads to an average of about three Munc13-1 clusters per given MF AZ (Figure 2C). This is slightly below the 5-6 Munc131 nano-assemblies detected per AZ with 3D STORM corresponding to release sites Sakamoto et al., 2018. Thus, we think that the nearest neighbor estimates (k1 type) as retrieved by ourprevious nearest neighbor gSTED analysis is a good representation of intra-AZ neighborhood relations, however, with a contribution of inter-AZ pairs as well (also see below).

To further appreciate these valid concerns, we now include data from previous and additional experiments using postsynaptic marker Homer1 as a reference for intra AZ clusters. As the postsynaptic density region is known to match the extent of the presynaptic active zone (planar membrane contact), by measuring distances between presynaptic clusters adjacent to one postsynaptic Homer1 cluster, we were able to more directly measure intra-AZ distances. We used this approach in particular in immunostaining experiments that measure Munc13-1 to Munc13-1 inter-clusters distances. For both WT and KO MF synapses, the inter-clusters distances retrieved by line profile measurement were somewhat smaller compared to nearest neighbor STED analysis. For example, at control animal MF synapses, the interMunc13-1 cluster distance retrieved with line profile measurements was (165 ± 3 nm, n=6) and with nearest neighbor analysis (213 ± 12 nm, n=6). The difference between the two methods may result from technical reasons or selection bias, but in any case, also line profile measurements showed that loss of RIM-BP2 resulted in a larger distance between two Munc13-1 clusters specifically at MF AZs (Figure 3—figure supplement 3). We are therefore now more confident that RIM-BP2 stabilizes Munc13-1 clusters specifically at MF AZs.

Also, line profile inter-cluster distances between two Cav2.1 clusters opposed to a single Homer1 cluster were comparable between WT and RIM-BP2 KO synapses (Figure 2—figure supplement 1) consistent with our previous nearest neighbor gSTED analysis.

Furthermore, line profile measurement between Munc13-1 to Cav2.1 clusters revealed that Munc13-1 clusters are app. 14 nm further away from Cav2.1 cluster in RIM-BP2 KO MF synapses (86 ± 4, n=6) compared to WT (72 ± 3 nm, n=8; p=0.022311, Mann Whitney U test) (Figure 3—figure supplement 1).

A) The NN-STED analysis is not very meaningful at MF terminals. While NN is insightful for methods that localize individual molecules, it is not for STED, which does not have this capacity. It is uncertain whether any protein cluster is within MF terminals or may be in other synapses around. The nature of biochemical interactions of these proteins suggests that they are intermixed as they all bind to one another, it is not clear that there are separable RIM or CaV2 or Munc13 clusters at distances of 200 nm of each other within an active zone. It is also not clear whether clusters of different proteins belong to the same active zone or even the same synapse. Some of these points are reflected in the data. In Figure 2, it is unclear how RIM, on average, ends up being closer to Homer1 than to CaVs, if the three belong to the same synapse (CaVs sit in the membrane that separates RIM and Homer). Figure 3 claims a reduction in Munc13 levels, but what the author's measure in their analysis is cluster numbers, not levels. It would be much preferable to have a measurement of Munc13 levels within a bouton or within an active zone. This could easily be done in experiments in cultured Dentate Gyrus neuron autapses. The authors should provide either the information on cluster analyzes or find a better way to analyze their STED images.

We thank the reviewers for raising these important points and we appreciate the chance to address them point by point.

1) Meaningfulness of NN gSTED analysis at MF synapses: We agree with the reviewers that our gSTED-based approach cannot localize individual molecules, however our method can detect areas enriched in AZ or postsynaptic proteins at sub-synaptic resolution. We termed these areas clusters, defined based on their fluorescence intensity, as described in our answer to major point 2. We are convinced that the gSTED analysis can inform about relative differences in the nanoscopic protein architecture of AZs based on our findings below.

Differentiating intra-AZ from inter-AZ clusters: As already addressed above, we agree that our nearest neighbor analysis did a priori not allow us to discriminate between clusters belonging to a single AZ or to nearby AZs. We therefore added an additional approach, please see reply to major point 2.

2) Average distance at which RIM1 cluster are found relative to Cav2.1 clusters: Our previous STED analysis at SC synapses showed that RIM1 is found at app. 134 nm from the postsynaptic marker Homer1 Grauel et al., 2016. These values are compatible with published data using STORMDani et al., 2010. Our previous study also showed that the nearest RIM1 cluster in average 185 nm apart from a given Cav2.1 cluster Grauel et al., 2016. The current study shows that the nearest RIM1 cluster is more distant from Cav2.1 clusters (254 nm), indicating structural differences between MF and SC synapses.

3) Munc13-1 levels: to address the reviewers’ question regarding Munc13-1 levels in the RIM-BP2 deficient neurons, we analyzed confocal images for Munc13-1 at MF and CA3-CA1 synapses in situ, acquired in parallel to the gSTED images. Indeed, we observe a significant reduction of Munc13-1 intensity levels specifically at MF synapses compared to CA3-CA1 synapses (Figure 3—figure supplement 3). We included these results in the revised manuscript.

B) On similar lines: One fundamental issue relating the overall conclusion is the number of clusters of each molecules and their spatial separations within the AZ. Given an average AZ area of 0.06 um2, the diameter of such average AZ (assuming disc shape) would be ~280 nm. The authors do not tell the number of clusters for each proteins, but their separation distances are around 260 and 110 nm. The reviewer would like to see the number of clusters per AZ for each presynaptic AZ molecules and their spatial arrangements that is compatible with the measured inter-cluster distances. Without knowing the numbers, the reviewer cannot comprehend whether such cluster separation distances can be consistent with the size of the AZ or not.

We fully concur with the concerns of the reviewers. And as addressed in the reply to major point 2, we now considered AZ size estimates in our calculations. With nearest neighbor gSTED analysis based on EM estimates of MF AZ size, we estimated three Munc13-1 cluster per given MF AZ, slightly below the 5-6 Munc13-1 nanoassemblies found with 3D STORM, corresponding to release sites Sakamoto et al., 2018. We further estimate four RIM-BP2, two Bassoon clusters, three Cav2.1 clusters and one RIM1 cluster per given MF AZ (Figure 2C,H). As second approach, to unequivocally determine intra-AZ distances, we turned to peak-to-peak line measurements of clusters apposed to a single postsynaptic density labeled by Homer1. This analysis indicates that MF AZs display at least two Cav2.1 and two Munc13-1 clusters per AZ, but yielded 20 – 30% shorter distances.

While we fully acknowledge that these measurements are rather proxies for the actual molecular architecture, they revealed whether distinct protein distributions change between wildtype and RIM-BP2 deficient synapses. We detected no change in the distance between two Cav2.1 clusters at MF AZ upon loss of RIM-BP2 (Figure 2—figure supplement 1), as previously shown with our nearest neighbor STED analysis. The inter-cluster distance for Munc13-1 however increased in mossy fiber AZs upon loss of RIM-BP2 (Figure 3—figure supplement 2).

C) How were distances between 'clusters' measured? For example, bassoon seems to form a homogeneous line in Supplementary Figure 1A and RIM-BP and Munc13-1 form some clusters. The authors should demonstrate on the figure how they measured these distances and they should provide more images from which the readers could have an impression of how similar or dissimilar the synapses were with respect to these proteins.Related to the problem of cluster distance quantification: The authors performed two sets of colocalization experiments: Cav2.1 vs. RIM1 and Munc13-1 vs. RIM BP2. The distances between these clusters were ~ 250 and 170 nm. What is the spatial relationships among them?

In general, we identified AZ protein clusters in deconvolved 2D gSTED images based on an uneven fluorescence signal distribution, upon thresholding and watershed segmentation. Per image, several hundreds to thousands of clusters were analyzed. The x and y coordinates of the center of mass for each cluster were retrieved by Amira software and Euclidean distances between clusters formed by two AZ proteins of interest were determined with a custom-written MATLAB script, previously published in Grauel et al., 2016. Values retrieved for each image were then averaged per mouse. We now show graphs indicating the average number of protein clusters found per each animal analyzed (Figure 2E, Figure 3C,D,I,J, Figure 3—figure supplement 4C,D,H,J).

We now included more detailed information about our nearest neighbor gSTED analysis in the Materials and methods section and cartoons to depict the analysis in each figure.

For each gSTED figure, we now included more insets from the images obtained to better show in details the synapses analyzed.

Regarding the spatial relationship of RIM1 vs. Munc13-1 or RIM-BP, unfortunately we cannot provide estimates. Due to limited availability of the anti RIM1 antibody, whose production has been discontinued by the producing company, we could not perform additional experiments to estimate RIM’s spatial relationship to RIM-BP2 clusters or to Munc13-1 clusters.

3) Another major shortcoming of the manuscript is the lack of detailed demonstration of how RIM-BP2 affects Schaffer collateral (SC) synapses. First, the authors did not perform all localization experiments in SC synapses in parallel with that of mossy fiber synapses. In addition, no functional experiment is presented that would test the prediction of the altered molecular composition of the SC synapses. Please provide the comparative functional data.

We thank the reviewers for pointing this out. In the previous manuscript, we merely compared our data from the MF synapses to previously published results at the SC synapse Grauel et al., 2016. Now, we added gSTED based experiments for RIM1/Cav2.1/Homer1 and Munc13-1/RIM-BP2/Bsn from MF synapses and SC synapses measured from the same brain sections and included these data in our manuscript (subsection “deletion does not alter Ca^2+^106 -channel localization at the MF synapse”).

Functional electrophysiological experiments at SC synapses in acute brain slices and in hippocampal glutamatergic neuron autaptic culture have been published by us Grauel et al., 2016. Briefly, we could show that at SC synapses loss of RIM-BP2 results in enhanced short term facilitation and that RIM-BP2 fine-tunes Cav2.1 voltage-gated Ca^2+^-channel localization at SC AZs.

4) A further important addition is to do a better paired-pulse ratio (PPR) analysis in cultured dentate gyrus neurons to assess probability of vesicular release (Pvr). The increased PPR strongly suggest a decrease in Pvr, but the analysis is limited to a single inter-spike-interval. The authors use the sucrose method as a measure of Pvr and argue that Pvr is unchanged, but it remains unclear whether this is a valid way to quantify Pvr at cultured dentate gyrus synapses. A complete analysis of PPRs at cultured dentate gyrus neurons and in hippocampal slices should be performed at various inter-stimulus intervals. This would better connect the slice and culture data sets, and could either support decreased Pvr, or reject this hypothesis. If PPRs indicate decreased Pvr, then this should be better stated in the Title and the Abstract.

Defining whether there is indeed a change in release probability turned out to be not trivial, and we acknowledge in the manuscript, that this remains to be clearly resolved. We applied the two major techniques to test whether release probability is altered upon RIM-BP2 deletion at MF synapses: PPR measurements and estimation of Pvr using hypertonic sucrose. Both methods have their advantages and disadvantages, but have been shown in the past to robustly correlate with each other at small synapses. However, it indeed might be that at MF synapses this relationship is not holding up. Therefore, we performed additional experiments:

a) In a first set of experiments we performed patch-clamp recordings from CA3-pyramidal cells in acute hippocampal slices and stimulated MF EPSCs – while the selectivity of mossy fiber inputs was always tested with DCGIV. In contrast to WT recordings, we rarely observed input of mossy fibers into area CA3 in the RIM-BP2 KO’s, most likely due to the strongly weakened synaptic output. However, this made it also unfeasible to properly quantify synaptic responses, and measures of release probability such as failure rate during minimal stimulation, or paired pulse behavior.

b) As suggested by the reviewer, we further characterized paired-pulse facilitation at RIM-BP2 KO granule autaptic neurons using, in addition to 25 ms, 5 additional inter-stimulus intervals (50ms, 100ms, 150ms, 250ms and 500ms). PPR differences were significant in all but the second longest interstimulus interval in granule autapses compared to WT (Figure 5G).

Together these data support the hypothesis that loss of RIM-BP2 at MF synapses reduces synaptic release probability. However, in contrast to SC synapses, in which we could correlate the decrease in release probability to alterations in Ca^2+^-channel localization, Cav2.1 localization and abundance is not significantly altered at MF active zones. Nevertheless, here we find that the distance at which Munc13-1 clusters are located relative to Ca^2+^-channels is significantly increased specifically at MF synapses in RIM-BP2 KO mice. This alteration in the distance between release sites, mapped by Munc13-1, and Ca^2+^-channels might account for the defect in release probability that we observed at MF terminals in absence of RIM-BP2. In addition, it might be that Munc13-2, since it’s not altered by the deletion of RIM-BP2 at MF terminals, contributes now more to vesicle priming and thus lowers the average release probability respectively Rosenmund et al., 2002. Determination of vesicular release probability using hypertonic sucrose solution however did not show a significant reduction, indicating that the sucrose based method lacks sensitivity to resolve a moderate reduction in vesicular release probability upon loss of RIM-BP2. Alternatively, the loss of RIM-BP2 has a more pronounced effects on processes that affect PPR but less so on Pvr, something that has been previously reported for Rab3 mutants Schluter et al., 2006.

We added the new data to the manuscript and revised the Discussion section about the differential impact of RIMBP2 deletion on short term plasticity and release probability.

[Editors' note: further revisions were requested prior to acceptance, as described below.]

Summary:After careful re-reading of the revised manuscript, all reviewers acknowledge the revision by the authors, the importance of the study and the fact that the study should be published at eLife. However, all reviewers agreed that the details of the image analysis and cluster analysis could be improved.1) The limitation being that it is unclear whether a cluster belongs to the same active zone or not. Therefore, please further discuss the details of the cluster analysis.

We agree with the reviewers that our nearest neighbor gSTED analysis doesn’t allow us to definitely discriminate between intra- and inter- active zone clusters. We, therefore, added in the Results section and Discussion section the following sentences, to further emphasize this limitation:

Subsection “M-BP2 deletion does not alter Ca^2+^106 -channel localization at the MF synapse”: “In order to define putative differences in the active zone architecture at MF and CA3-CA1 synapses, we utilized super-resolution STED-microscopy based on detection of major active zone proteins intensities, here referred to as protein clusters. Quantification of the localization of these protein clusters was mainly performed by counting and measuring distances between intensities of these marker proteins.”

Subsection “M-BP2 deletion does not alter Ca^2+^106 -channel localization at the MF synapse”: “Our gSTED analysis does not allow the differentiation between intra- and inter- active zone protein clusters. However, ultrastructural quantifications of MF active zone size (0.12 µm2)^1^, are consistent with four RIM-BP2, two Bassoon, and three Munc13-1 clusters per active zone (Figure 2c). Regardless of this semiquantitative analysis, the difference in Munc13-1/RBP cluster distances is indicative for distinct active zone organization between MF and CA3-CA1 synapses.”

Subsection “M-BP2 deletion does not alter Ca^2+^106 -channel localization at the MF synapse”: “However, it does not allow us to discriminate protein clusters within or between nearby active zones unequivocally.”

Discussion section: “When comparing apparent distances between active zone protein clusters, clusters of the presynaptic scaffold proteins RIM-BP2 and Munc13-1 were found at larger distances at MF synapses as compared to SC synapses. Also, the distance of the closest neighboring cluster of RIM1 to a given Cav2.1 cluster is larger compared to the distance we previously found at SC synapses^2^. Therefore, our STED analysis suggests that a single active zone may contain more than one protein cluster of key active zone proteins. Some of these key players might have a differential relative localization between each other at MF synapses in comparison to SC synapses. However, further evidence of sub-active zone organization is required to verify this hypothesis. Moreover, more sophisticated methods of defining protein clusters to belong to the same active zone will be helpful in defining the architecture of the active zone of mammalian central synapses.”

2) In this context, please tone down the importance of the cluster analysis in the Abstract.

In the old version of the manuscript, we stated:

“Interestingly, in wild type mossy fiber synapses, the distance between RIM-BP2 clusters and Munc13-1 clusters is larger than in hippocampal pyramidal CA3-CA1 synapses, suggesting that spatial organization may dictate the role a protein plays in synaptic transmission and that differences in active zone architecture is a major determinant factor in the functional diversity of synapses.”

This was changed to:

“Differences in the active zone organization may dictate the role a protein plays in synaptic transmission and that differences in active zone architecture is a major determinant factor in the functional diversity of synapses.”

We hope that the reviewers agree with our changes.

3) Moreover, emphasize the problem with the availability of the RIM antibody that readers understand the limitations.

We now inserted the following sentences in subsection “Immunohistochemistry, time gated STED microscopy and image analysis”:

“Notably, the antibody against RIM1 (RRID: AB_2315284, BD Pharmigen) used in this study is not commercially available anymore. To repeat these results, other RIM1 antibodies need to be utilized and validated beforehand.”

ELife – –4) Subsection “RIM-BP2 docks synaptic vesicles via the specific recruitment of Munc13-1 at MF synapses”: the docking deficit is ~25%, not 50%, and the text refers to the wrong figure panel.

We changed the percentage accordingly.